# PD-1 pathway regulates ILC2 metabolism and PD-1 agonist treatment ameliorates airway hyperreactivity

Doumet Georges Helou [1], Pedram Shafiei-Jahani[1,4], Richard Lo[1,4], Emily Howard[1], Benjamin P. Hurrell[1], Lauriane Galle-Treger[1], Jacob D. Painter [1], Gavin Lewis[2], Pejman Soroosh[2], Arlene H. Sharpe [3] & Omid Akbari [1]✉

Allergic asthma is a leading chronic disease associated with airway hyperreactivity (AHR). Type-2 innate lymphoid cells (ILC2s) are a potent source of T-helper 2 (Th2) cytokines that promote AHR and lung inflammation. As the programmed cell death protein-1 (PD-1) inhibitory axis regulates a variety of immune responses, here we investigate PD-1 function in pulmonary ILC2s during IL-33-induced airway inflammation. PD-1 limits the viability of ILC2s and downregulates their effector functions. Additionally, PD-1 deficiency shifts ILC2 metabolism toward glycolysis, glutaminolysis and methionine catabolism. PD-1 thus acts as a metabolic checkpoint in ILC2s, affecting cellular activation and proliferation. As the blockade of PD-1 exacerbates AHR, we also develop a human PD-1 agonist and show that it can ameliorate AHR and suppresses lung inflammation in a humanized mouse model. Together, these results highlight the importance of PD-1 agonistic treatment in allergic asthma and underscore its therapeutic potential.

[1] Department of Molecular Microbiology and Immunology, Keck School of Medicine, University of Southern California, Los Angeles, CA, USA. [2] Janssen Research and Development, San Diego, CA, USA. [3] Department of Immunology, Harvard Medical School, Boston, MA, USA. [4] These authors contributed equally: Pedram Shafiei-Jahani, Richard Lo. ✉email: akbari@usc.edu

Allergic asthma is a chronic inflammatory airway disease associated with respiratory symptoms, such as shortness of breath, chest tightness, and cough[1]. The prevalence of asthma has increased considerably in recent decades to affect ~339 million people worldwide[2]. Therefore, the identification of novel therapeutic targets has become a necessity to ensure efficient control of this disease. Animal models are useful and can closely mimic clinical features of asthma, mainly airway hyper-reactivity (AHR), which relies on T-helper 2 (Th2) cytokine secretion; interleukin (IL)-5 promotes eosinophil recruitment to the lungs, while IL-13 induces goblet cell hyperplasia and mucus production[3]. Although Th2 cells were classically recognized as key controllers of type 2 inflammation, the early activation of type-2 innate lymphoid cells (ILC2s) has emerged as a critical step in the initiation and amplification of asthma[4–6].

ILC2s are a subset of the innate lymphoid cell family that mirror Th2 cells in many ways. Functionally, activated ILC2s secrete high amounts of type-2 cytokines, such as IL-4, IL-5, IL-9, and IL-13, in response to non-specific alarmins that include IL-25, IL-33, and thymic stromal lymphopoietin (TSLP)[7]. ILC2s are thus highly implicated in various diseases, such as allergic rhinitis, atopic dermatitis and allergic asthma[8]. Phenotypically, ILC2s are lineage negative and do not express the rearranged antigen receptors; however, ILC2s do express multiple co-stimulatory molecules that are critical for the adaptive immune response[9,10]. Among those molecules, ILC2s express the inducible T-cell co-stimulator (ICOS), ICOS ligand (ICOS-L)[11], death receptor 3 (DR3)[12], tumor necrosis factor receptor 2 (TNFR2)[13], and glucocorticoid-induced tumor necrosis factor receptor (GITR)[14]. Similar to T cells, ILC2s also express programmed cell death protein-1 (PD-1)[15–17], a potent immune inhibitory receptor.

PD-1 is a member of the B7/CD28 family and is mainly expressed on activated T cells, B cells, and macrophages. Binding of PD-1 to its ligands, PD-L1 and PD-L2, is followed by a cascade of intracellular signaling that results in T cell inhibition and exhaustion[18,19]. Briefly, the tyrosine residues in PD-1 cytoplasmic domain undergo phosphorylation, promoting recruitment of protein tyrosine phosphatases (PTPs), mainly Src homology region 2 domain-containing phosphatase-2 (SHP-2). This subsequently leads to the dephosphorylation of several kinases that are necessary for cell survival and proliferation[20–23]. Blocking the PD-1 pathway has become a therapeutic strategy for multiple tumor types, given its capacity to restore T cell functions[24]. There are currently six PD-1 pathway inhibitors approved by the Food and Drug Administration (FDA) for 16 tumor types[25]. Conversely, PD-1 deficiency is associated with altered self-tolerance and autoimmune disorders in mice. Therefore, the PD-1 axis is now gaining great attention as a potential therapeutic target for the treatment of various inflammatory and autoimmune diseases[26,27].

In this study, we evaluate the protective function of PD-1 on the development of AHR by focusing on its capacity to regulate ILC2 activation. PD-1 is highly inducible in pulmonary ILC2s and shapes their transcriptional activity, activation, and metabolism. Using a humanized mouse model, we show that a PD-1 agonist could be a potential therapeutic for allergic asthma, as it represses ILC2 proliferation and reduces ILC2-mediated secretion of Th2 cytokines. Our findings provide new insights into the mechanisms of ILC2 regulation, which might be used to develop new approaches for the treatment of allergic asthma.

## Results

**PD-1 is highly inducible in IL-33-activated ILC2s.** While PD-1 expression is usually associated with T cell exhaustion, PD-1 is expressed on activated cells that include different subsets of T cells, B cells, and natural killer cells[28]. We first characterized the expression of PD-1 on pulmonary ILC2s, which were identified by flow cytometry as CD45[+] lineage[-] ST2[+] CD127[+] (Fig. 1a, Supplementary Fig. 1A). Throughout the study, ILC2s from unchallenged mice were defined as naïve (nILC2s), while ILC2s from IL-33-challenged mice were defined as activated (aILC2s). PD-1 was expressed only in a small population of nILC2s; however, intranasal stimulation with IL-33 resulted in a remarkable induction of PD-1 expression (Fig. 1b, Supplementary Fig. 1B). We also assessed PD-L1 and PD-L2 expression on pulmonary ILC2s. PD-L1 expression was high in both naïve and aILC2s, while PD-L2 was not expressed (Fig. 1c, d). The interaction between PD-1 and its ligands is required for the generation and delivery of inhibitory signals. To identify the potential sources of PD-1 ligands in our ILC2-dependent asthma model, we characterized the expression of PD-L1 and PD-L2 on CD45[+] and CD45[−] live pulmonary cells (Fig. 1e). Two positive populations were identified within the CD45[+] cells: (i) a population simultaneously expressing PD-L1 and PD-L2 that primarily represents CD11b[+] CD11c[+] cells, (ii) a population expressing only PD-L1 that mostly represents CD11b[+] Gr-1[+] cells (Fig. 1f). Interestingly, the percentage of these two populations increased significantly in response to IL-33 induction (Fig. 1g, h). In parallel, a population of CD45[−] cells was able to express and upregulate the expression of PD-L1 in response to IL-33 (Fig. 1i, j). Altogether, these results indicate that PD-1 is highly inducible on ILC2s and that immune and non-immune lung populations can provide a source of PD-1 ligands in the context of ILC2-dependent asthma.

**PD-1 regulates cytokine production and survival in aILC2s.** Given that PD-1 is remarkably expressed in pulmonary aILC2s, we compared the transcriptional profile of FACS-sorted aILC2s from wild-type (WT) and PD-1-knockout (KO) mice using RNA-sequencing analysis. Interestingly, the absence of PD-1 resulted in 840 significantly modulated genes in aILC2s: 426 genes were upregulated, while 414 were downregulated (Fig. 2a, Supplementary Table 1). To elucidate the capacity of PD-1 to regulate ILC2 effector functions, we first compared the expression of cytokine-encoding genes in WT and PD-1 KO aILC2. This resulted in a list of significantly different genes encoding Th2 cytokines, including *Il5*, *Il13*, *Il9*, and *Csf2*, that are highly upregulated in the absence of PD-1 (Fig. 2b). These observations were next confirmed at the protein level. aILC2s from WT and PD-1 KO mice were cultured for 24 h to quantify cytokines in the culture supernatant. As expected, ILC2 activation resulted in Th2 cytokine secretion; however, PD-1 KO aILC2s exhibited higher amounts of IL-5, IL-13, and IL-9 (Fig. 2c–e). To assess the direct impact of PD-1 crosslinking on cytokine production, we first determined whether PD-L2 would inhibit Th2 cytokine production. Consistent with our previous experiments, we observed that PD-L2 Fc decreased the secretion of IL-5 and IL-13 by about 2- to 3-fold (Fig. 2f, g). We then examined whether the constitutive expression of PD-L1 could represent a source of functional PD-1 ligand in ILC2s. To investigate this hypothesis, nILC2s were stimulated in vitro with IL-33 in the absence of any other source of PD-1 ligands, as well as in the presence of PD-1 blocking antibody or the isotype control. Interestingly, the expression of the key transcription factor GATA-3, as well as the production of IL-5, increased significantly upon PD-1 blockade in WT ILC2s (Fig. 2h, i). These results suggested the inhibition of a potential PD-1/PD-L1 interaction in ILC2s. Moreover, in vitro activated PD-1 KO ILC2s displayed a significantly different phenotype as compared to WT ILC2s (Supplementary Fig. 2A–D). Taken together, these results suggest that PD-1 interaction with PD-L2

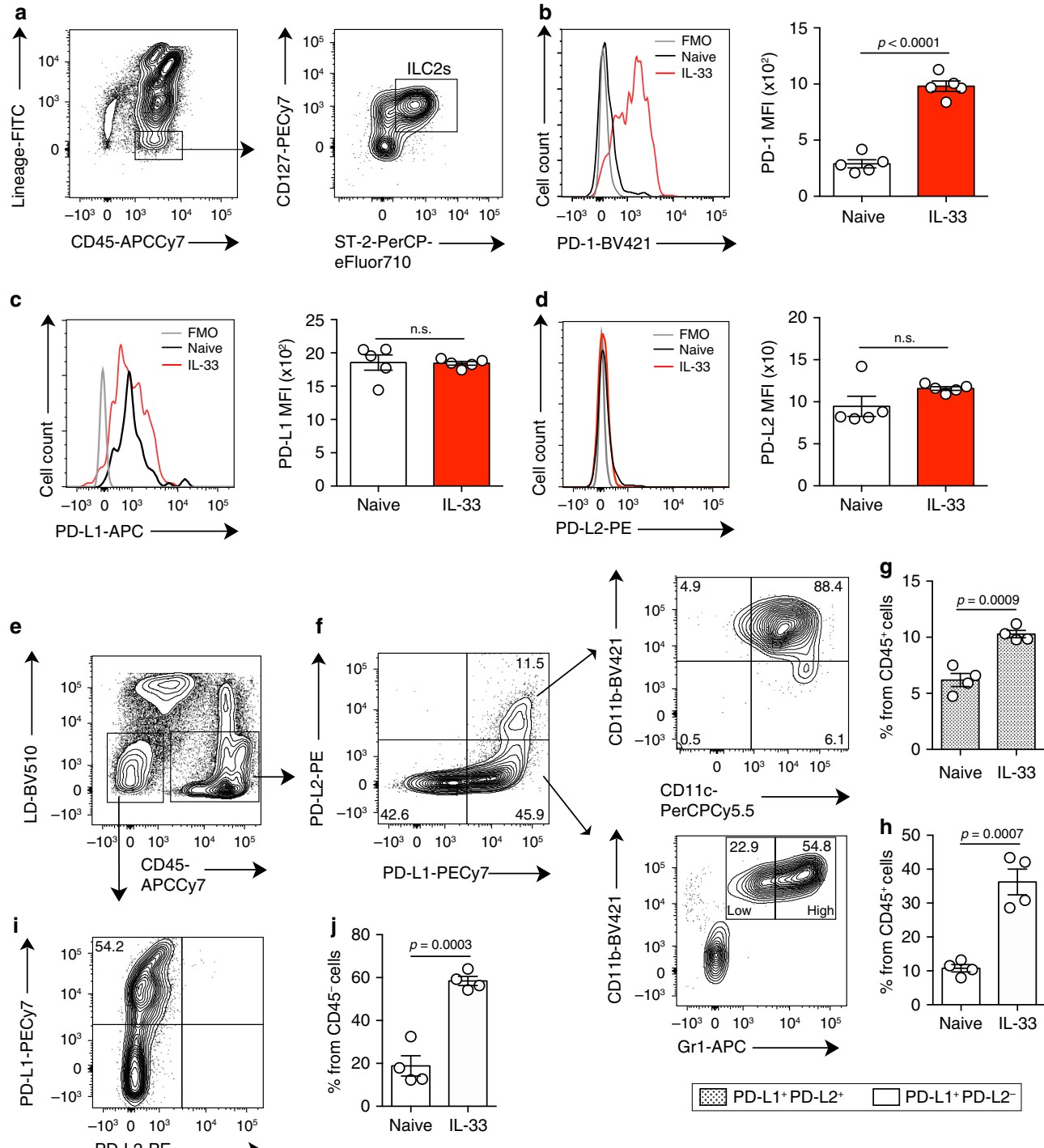

**Fig. 1 IL-33 induces PD-1 expression in pulmonary ILC2s. a** Pulmonary ILC2s were defined and/or sorted by a lack of lineage markers (CD3e, CD45R, Gr-1, CD11c, CD11b, Ter119, CD5, TCR-β, TCR-γδ, NK1.1, and FcεRI) and expression of CD45, ST2, and CD127. **b–j** BALB/cByJ mice (WT) mice were challenged or not (naïve) with 0.5 μg of rm-IL-33 for 3 consecutive days. ILC2s from IL-33-challenged mice were defined as activated (aILC2s). **b** Representative histogram of the expression of PD-1, **c** PDL-1, and **d** PDL-2 in pulmonary ILC2s and corresponding quantification (right) presented as Mean Fluorescence Intensity (MFI); $n = 5$. **e** Gating strategy to define PD-L1 and PD-L2 live positive cells from **f** CD45+ and **i** CD45− cells in IL-33 challenged mice. **g, h** Percentage of PD-L2+ and/or PD-L1+ cells from CD45+ and **j** CD45− cells; $n = 4$. Data are representative of three independent experiments and are presented as means ± SEM (two-tailed Student's $t$ test, n.s. non-significant).

negatively regulates cytokine production and that PD-L1 expression on ILC2s plays a functional role in their regulation.

PD-1 was first discovered as a cell death inducer[29]. Therefore, the role of PD-1 axis in pulmonary ILC2 survival was investigated in this study at different levels. At the transcriptional level,

proapoptotic genes, such as *Bid* and *Casp2*, were significantly downregulated in PD-1 KO compared to WT aILC2s, while many anti-apoptotic genes, including *Bcl2l1*, *Bag3*, and *Mcl1*, were upregulated (Fig. 2j). To confirm the involvement of PD-1 in ILC2 survival, we used the AnnexinV apoptotic marker and the

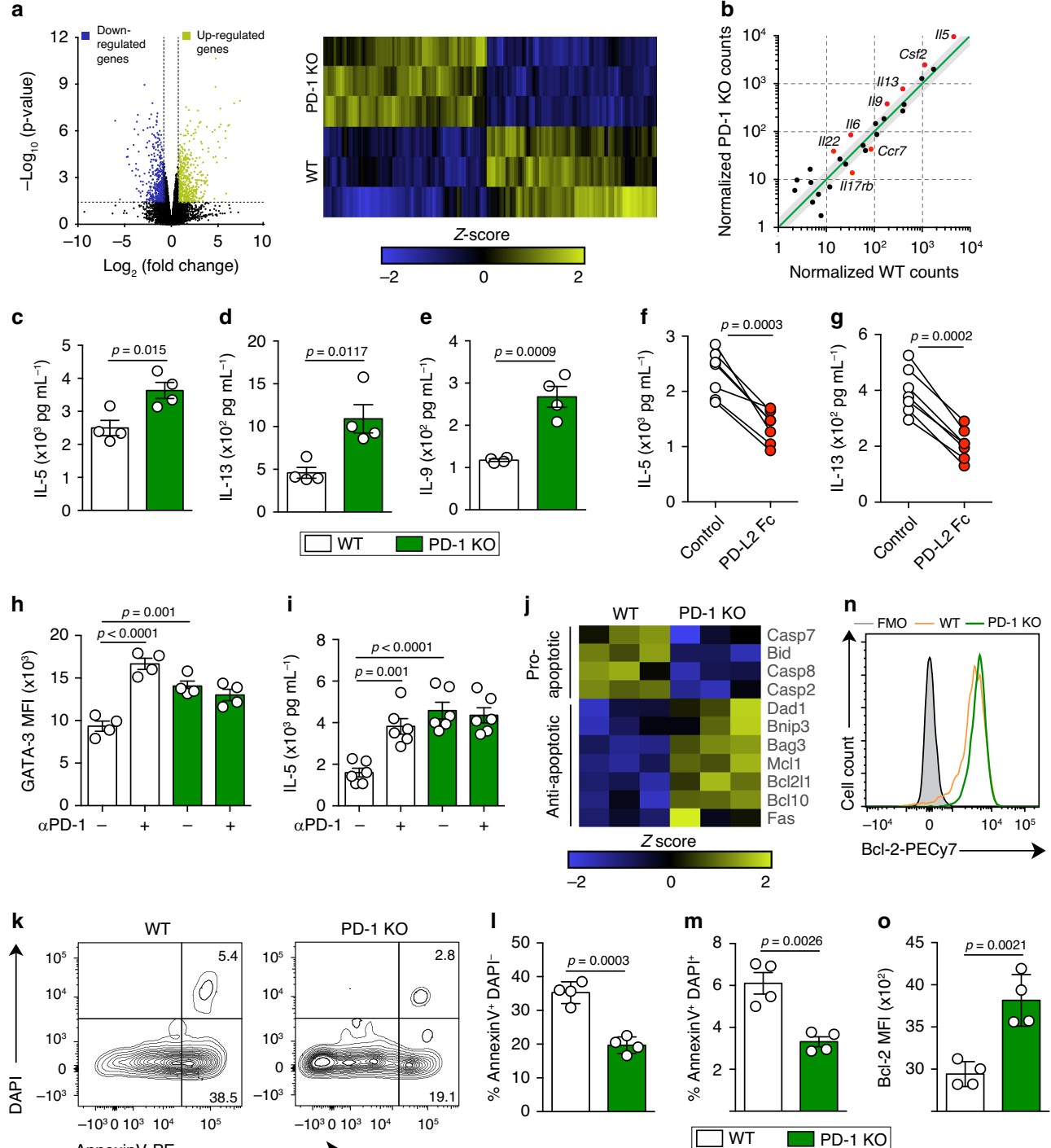

**Fig. 2 PD-1 axis controls cytokine production by aILC2s and decreases their survival. a–g; j–o** ILC2s were sorted from WT and PD-1 KO mice after three intranasal challenges with 0.5 µg of rm-IL-33. Sorted cells were incubated with rm-IL-2 (10 ng mL$^{-1}$) and rm-IL-7 (10 ng mL$^{-1}$) for 24 h. **a** Volcano plot comparison (left) and heatmap representation (right) of total differentially regulated genes. Gene−specific analysis (GSA) algorithm was used to test for differential expression of genes ($p$-value < 0.05, $n = 3$). **b** Differentially expressed cytokine and cytokine receptor genes plotted as the normalized counts in WT compared to PD-1 KO aILC2s. Gray area indicates region of 1.5-fold-change cutoff or lower change in expression; $n = 3$. **c** Levels of IL-5, **d** IL-13, and **e** IL-9 quantified using LEGENDplex bead-based immunoassay; $n = 4$. **f** Levels of IL-5 and **g** IL-13 secreted by aILC2s in the presence of PD-L2 Fc or the isotype; $n = 7$. **h, i** ILC2s were sorted from naïve mice (nILC2s), cultured and stimulated in vitro with rm-IL-33 (20 ng mL$^{-1}$) for 48 h. PD-1 blocking antibody (αPD-1; 10 µg mL$^{-1}$) or isotype control were added in some conditions. **h** GATA-3 quantification presented as MFI in WT and PD-1 KO ILC2s; $n = 4$. **i** Levels of IL-5 quantified in ILC2 supernatants; $n = 6$. **j** Heatmap representation of significantly regulated pro- and anti-apoptotic genes. GSA algorithm was used to test for differential expression of genes ($p$-value < 0.05, $n = 3$). **k** Representative flow cytometry plots of AnnexinV DAPI staining and **l** corresponding quantification presented as the percentage of apoptotic and **m** dead ILC2s; $n = 4$. **n** Representative flow cytometry plot of Bcl-2 staining and **o** corresponding quantification presented as MFI in WT and PD-1 KO aILC2s; $n = 4$. Data are representative of three independent experiments and are presented as means ± SEM (two-tailed Student's $t$ test or one-way ANOVA).

dead cell discrimination dye, DAPI. Both AnnexinV$^+$ DAPI$^-$ and double positive cells were significantly decreased in PD-1 KO aILC2s, suggesting a decrease in cell apoptosis and death (Fig. 2k–m). Lastly, we assessed the expression of the apoptosis suppressor protein Bcl-2, using flow cytometry. As expected, Bcl-2 expression was significantly and consistently higher in PD-1 KO aILC2s compared to WT aILC2s (Fig. 2n, o). Altogether, these results indicate that the PD-1 axis actively participates in the regulation of ILC2 survival.

**The lack of PD-1 enhances glycolytic metabolism in aILC2s.** Aerobic glycolysis is considered a hallmark of T cell activation and proliferation[30]. While amino acids and aerobic glycolysis may be essential for ILC2s[31], ILC2 metabolism appears to be more dependent on fatty acids (FA)[32–34]. To investigate the impact of PD-1 on aILC2 metabolism, we examined the expression of several genes known to be highly implicated in these physiological mechanisms. Surprisingly, PD-1 deficiency led to a significant upregulation of glycolytic genes (such as *Hk1*, *Pkm*, and *G6pdx*) and a downregulation of those implicated in FA metabolism. Glutaminolysis-dependent genes were also upregulated in PD-1 KO aILC2s (Fig. 3a). To confirm the effect of PD-1 on glycolysis, we used the fluorescent glucose analog, 2-NBDG, and flow cytometry to measure the extent of glucose incorporation in living cells. In line with transcriptional analysis, the lack of PD-1 in aILC2s increased glucose uptake, suggesting for a differential glucose metabolism and an increased glucose consumption (Fig. 3b). Moreover, the expression of the main glucose transporter, Glut-1, was upregulated in PD-1 KO ILC2s as compared to WT ILC2s in response to IL-33 (Supplementary Fig. 2B). These differences were not observed at steady state in freshly sorted nILC2s (Supplementary Fig. 2E, F). Metabolomics was further used to assess the impact of PD-1 deficiency on metabolites. Levels of metabolites from glycolysis and the alternative pentose phosphate pathway were significantly elevated in PD-1 KO aILC2s when compared to WT aILC2s. These metabolites include phosphoglyceric acid, UDP-Glucose, pentose phosphate, and hexose phosphate (Fig. 3c–f). To confirm that differential 2-NBDG uptake and metabolomics results reflect functional differences in glycolytic metabolism, we measured oxygen consumption rate (OCR), reflective of active oxidative phosphorylation (OXPHOS), and extra cellular acidification rate (ECAR). PD-1 KO aILC2s exhibited a decrease in spare respiratory capacity, demonstrating that the lack of PD-1 triggers the switch from OXPHOS to aerobic glycolysis (Fig. 3g). Under these circumstances, glucose is fermented into lactate instead of being oxidized in mitochondria due to a high energy requirement. Additionally, cell energy phenotype profiling showed greater ECAR in PD-1 KO than WT aILC2s, thus implicating a lower OCR/ECAR ratio at maximal respiration and a higher glycolytic capacity (Fig. 3h, i). These results collectively reveal that PD-1 deficiency is associated with increased glucose demand in aILC2s, driving a significant metabolic shift towards glycolysis.

**PD-1 inhibits methionine and glutamine catabolism in aILC2s.** Amino acid catabolism generates metabolites for protein synthesis but can also act as signaling molecules to control immune cell growth, nucleotide synthesis, redox control, and many other functions[35]. Our metabolomic analysis showed that the lack of PD-1 generally altered the metabolic activity of aILC2s. In particular, there was an upregulation of intermediate metabolites that are implicated in pyrimidine and purine synthesis, such as adenosine 5′-diphosphate (ADP) and adenine, as well as in methylation (Fig. 4a). Further analysis allowed for the characterization

of two specific pathways that are greatly affected in the absence of PD-1. The first pathway is that of methionine (Fig. 4b). The relative amount of two major intermediate metabolites in methionine catabolism, S-Adenosylmethionine (SAM) and S-Adenosylhomocysteine (SAH), were significantly increased in PD-1 KO aILC2s (Fig. 4c, d). Furthermore, two end metabolites of methionine catabolism, glutathione and taurine, exhibited higher relative amounts in PD-1 KO aILC2s as compared to WT aILC2s (Fig. 4e, f). This suggests that PD-1 expression limits the metabolism of methionine in aILC2s.

Among the screened pathways, glutaminolysis was also affected (Fig. 4g). Glutamine consumption is critical in immune cell metabolism, particularly for lymphocyte proliferation and cytokine production. In addition, glutaminolysis provides the nitrogen donor in the formation of nucleic acids[36,37]. Our results revealed that the lack of PD-1 increased the relative levels of several intermediates and end metabolites involved in glutamine metabolism such as glutamic acid, *N*-methyl glutamate, *N*-acetyl glutamic acid (NAG), and ornithine (Fig. 4h–k). Only the relative amount of GABA was significantly lower in PD-1 KO as compared to WT aILC2s (Fig. 4l). Taken together, these results suggest that PD-1 deficiency enhances amino acid catabolism in aILC2s, particularly in methionine- and glutamine-dependent metabolic pathways.

**PD-1 limits aILC2 proliferation via metabolic regulation.** The increase in metabolic activity is a hallmark of cell proliferation and protein synthesis. As Th2 differentiation and proliferation are tightly associated with GATA-3 and signal transducer and activator of transcription 5 (STAT5) activation[38], we tested whether PD-1 deficiency could alter the expression of some relevant transcription factors in aILC2s. We observed that PD-1 deficiency upregulated genes involved in Th2 differentiation signaling pathways, including *Gata-3*, *Junb*, *Stat5a*, and *Stat6*, and downregulated those involved in Th1 differentiation and activation, such as *Irf1* and *Stat1* (Fig. 5a). In line with these results, the intensity of GATA-3 expression was higher in aILC2s lacking PD-1, as assessed by flow cytometry (Fig. 5b). In addition, Ki67 intranuclear staining revealed that live aILC2s become highly proliferative in the absence of PD-1 (Fig. 5c). To assess the causal relationship between the metabolic shift in PD-1 KO ILC2s and proliferation, we inhibited glycolysis and methionine catabolism using the competitive inhibitors 2-deoxy-d-glucose (2-DG) and cycloleucine (CYL), respectively. The 2-DG forms the 2-DG-6-P that cannot undergo further glycolysis, while the CYL inhibits the methionine adenosyl transferase (MAT) enzyme that catalyzes the transformation of methionine into SAM (Fig. 5d). Relatively low concentrations of these inhibitors were sufficient in significantly decreasing GATA-3 expression and proliferation in PD-1 KO aILC2s, while the effect in WT aILC2s was weak or absent (Fig. 5e, f). To put our previous observations into context, we studied the effect of in vivo glycolysis inhibition on aILC2s. The inhibitor 2-DG was injected intraperitoneally along with intranasal administrations of IL-33 as described in Fig. 5g. PD-1 KO mice displayed a higher number of pulmonary ILC2s as compared to WT mice. Interestingly, 2-DG significantly decreased ILC2 percentage (from CD45$^+$, lineage$^-$ cells) and absolute count in PD-1 KO mice lungs, but not in WT mice (Fig. 5h, i). In line with these results, 2-DG treatment led to a significant decrease in GATA-3 and Ki67 expressions in PD-1 KO pulmonary aILC2s, while very weakly affecting WT aILC2s (Fig. 5j, k). Taken together, both ex vivo and in vivo experiments strongly suggest that PD-1 deficiency is associated with a metabolic shift toward glycolysis, thus enhancing ILC2 activation and proliferative potential in IL-33-induced asthma.

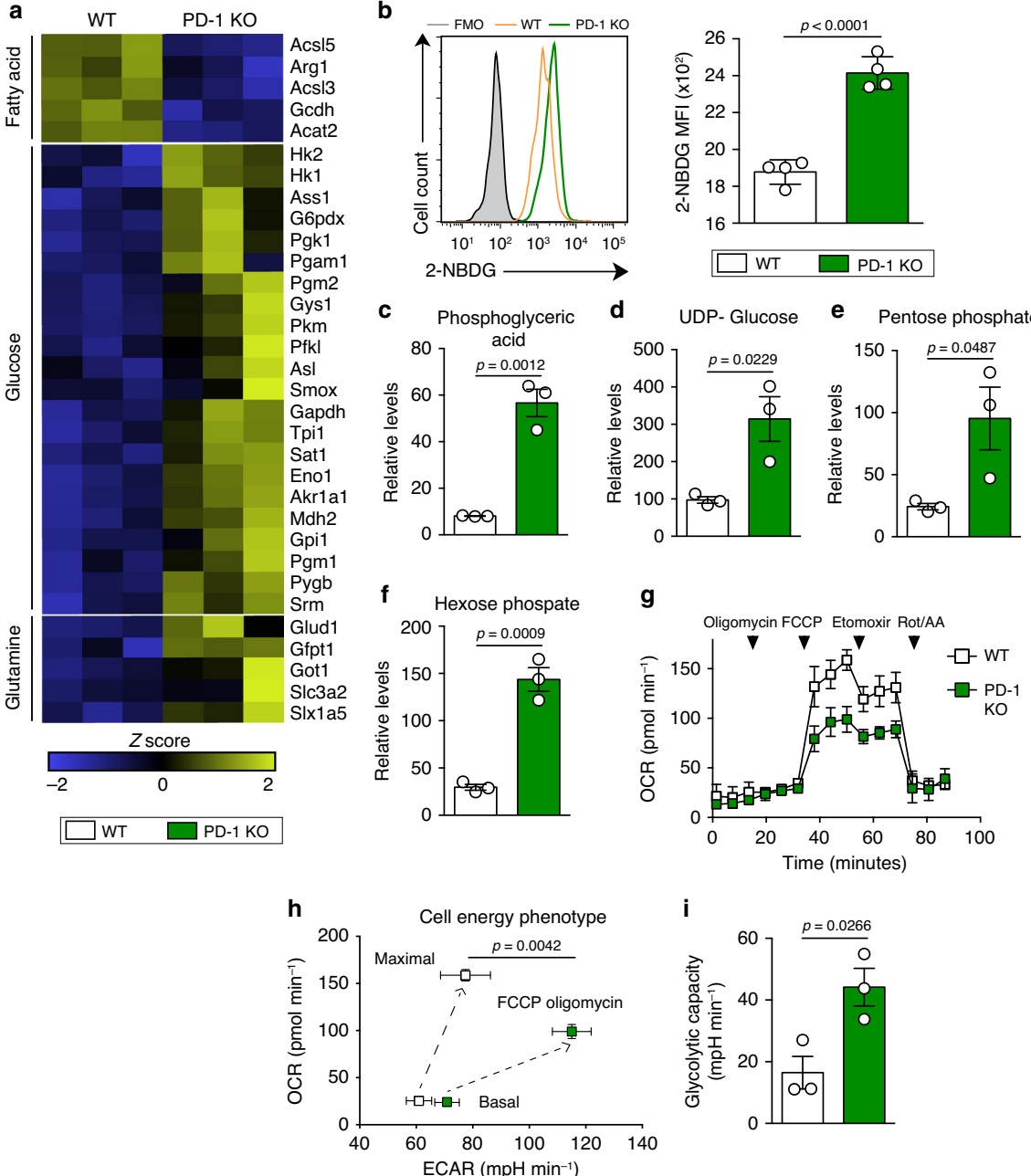

**Fig. 3 The lack of PD-1 signaling upregulates glycolysis in aILC2s. a–i** ILC2s were sorted from WT and PD-1 KO mice after three intranasal challenges with 0.5 μg of rm-IL-33. Sorted cells were incubated with rm-IL-2 (10 ng mL$^{-1}$) and rm-IL-7 (10 ng mL$^{-1}$) for 24 h. **a** Heat map representation of differentially regulated genes involved in the metabolism of aILC2 from WT and PD-1 KO mice. GSA algorithm was used to test for differential expression of genes (*p*-value < 0.05, *n* = 3). **b** Representative histogram (left) of 2-NBDG uptake and the corresponding quantification (right) presented as MFI in FACS-sorted WT and PD-1 KO aILC2s; *n* = 4. **c–f** Relative levels of metabolites in the glycolysis pathway analyzed using an LC-MS/MS system. **g** Measurement of the Oxygen consumption rate (OCR) under basal conditions and in response to indicated drugs. **h** Cell energy phenotype presented as OCR against extracellular acidification rate (ECAR). **i** Glycolytic capacity calculated as the difference between maximal and basal ECAR. Data are presented as means ± SEM (*n* = 3; two-tailed Student's *t* test).

**PD-1 reduces ILC2-mediated AHR and lung inflammation.**
Having demonstrated that PD-1 controls pulmonary ILC2 transcriptional profile, cytokine production, and limits ILC2 proliferation through metabolic regulation, we consequently hypothesized that PD-1 may contribute to the regulation of AHR and lung inflammation. WT and PD-1 KO mice received intranasal administrations of either IL-33 or PBS for 3 consecutive days. One day after the last intranasal challenge, lung function was assessed by direct measurement of lung resistance and

dynamic compliance (cDyn) in anesthetized tracheostomized mice using the FinePointe RC system (Buxco Research Systems), followed by bronchoalveolar lavage (BAL) and lung tissue sample analysis (Fig. 6a). As expected, intranasal administration of IL-33 significantly increased lung resistance in WT and PD-1 KO mice; however, lung resistance in IL-33-treated PD-1 KO mice was significantly higher than in IL-33-treated WT mice (Fig. 6b). In agreement with lung resistance, results of dynamic compliance showed lower response in IL-33-treated PD-1 KO compared to

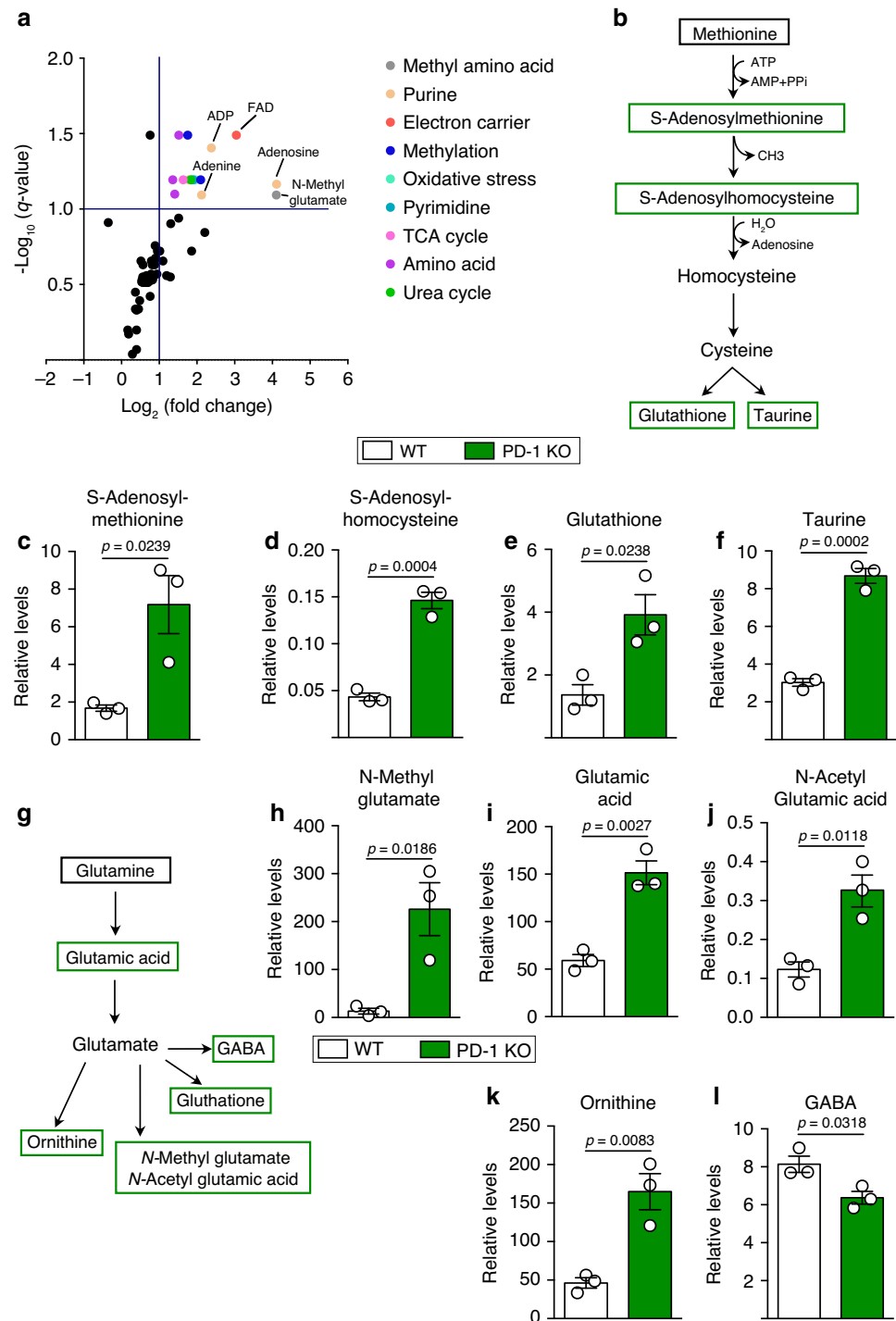

**Fig. 4 Lack of PD-1 enhances methionine and glutamine catabolism in pulmonary aILC2s. a–l** ILC2s were sorted from WT and PD-1 KO mice after three intranasal challenges with 0.5 µg of rm-IL-33. Sorted cells were incubated with rm-IL-2 (10 ng mL⁻¹) and rm-IL-7 (10 ng mL⁻¹) for 24 h. **a** Volcano plot comparison of the relative levels of cellular metabolites, analyzed using an LC-MS/MS system. Color annotation were attributed to differentially abundant metabolites according to their classification (2-fold change cutoff, q-value ≥ 0.1; $n = 3$). **b** Methionine catabolism pathway showing measured metabolites in green. **c** Relative levels of measured methionine catabolism metabolites: SAM, **d** SAH, **e** Glutathione, and **f** Taurine in WT and PD-1 KO aILC2s. **g** Glutaminolysis pathway showing measured metabolites in green. **h** Relative levels of measured glutaminolysis metabolites: N-Methyl glutamate, **i** Glutamic acid, **j** N-Acetyl Glutamic acid, **k** Ornithine, and **l** GABA in WT and PD-1 KO aILC2s. Data are presented as means ± SEM ($n = 3$; two-tailed Student's t test).

IL-33-treated WT mice (Fig. 6c). This indicated that PD-1 is a negative regulator in IL-33-induced AHR. Although IL-33 treatment significantly induced eosinophilia in the BAL of WT and PD-1 KO mice, the number of eosinophils was significantly higher in IL-33-treated PD-1 KO mice compared to WT mice

(Fig. 6d). Similarly, the number of ILC2s was higher in PD-1 KO mice lungs following IL-33 stimulation and even in naïve mice, as was the number of IL-5⁺ IL-13⁺ ILC2s (Fig. 6e, f). Given that PD-1 was not reported to affect ILC2 development[17], the observed differences at steady state could be due to PD-1

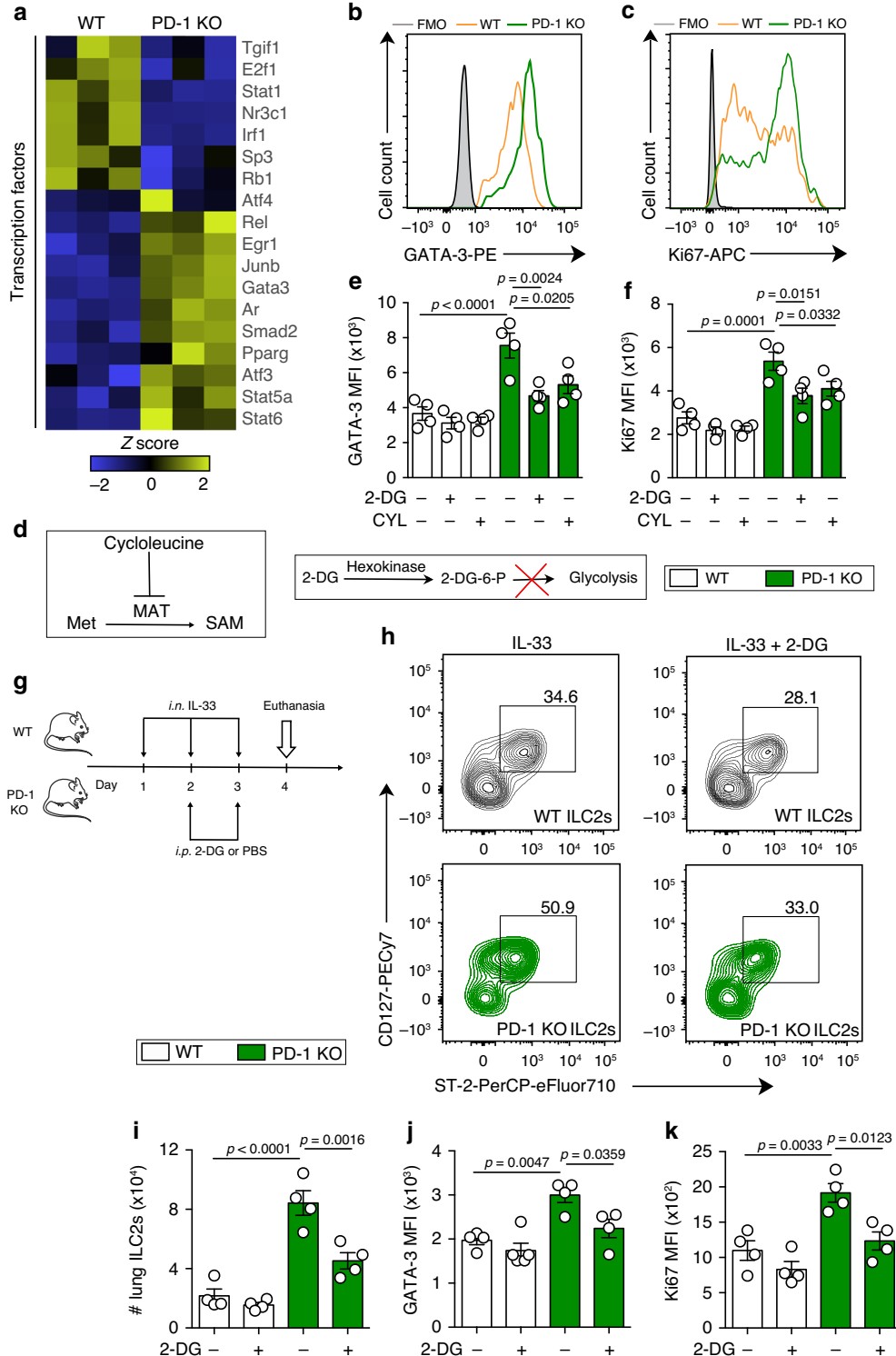

**Fig. 5 PD-1 controls aILC2 proliferation through metabolic regulation. a**–**f** ILC2s were sorted from WT and PD-1 KO mice after three intranasal challenges with 0.5 μg of rm-IL-33. Sorted cells were incubated with rm-IL-2 (10 ng mL$^{-1}$) and rm-IL-7 (10 ng mL$^{-1}$) for 24 h. **a** Heat map representation of differentially regulated transcription factors in FACS-sorted aILC2s from WT and PD-1 KO mice lungs. GSA algorithm was used to test for differential expression of genes (*p*-value < 0.05, *n* = 3). **b**–**f** Sorted aILC2s were cultured in vitro for 24 h with 2-DG (0.5 mM) or CYL (20 mM) in some conditions. **b** Representative flow cytometry plots of GATA-3 and **c** intranuclear Ki67 with **e**, **f** the respective corresponding quantification presented as MFI in live WT and PD-1 KO aILC2s. **g**–**k** WT and PD-1 KO mice were intranasally challenged for 3 consecutive days with 0.5 μg rm-IL-33. On day 2 and 3, mice were intraperitoneally (i.p.) injected with 2-DG (500 mg kg$^{-1}$). On day 4, mice were euthanized. **h** Representative flow cytometry plots of ILC2s gated as CD127$^+$ ST-2$^+$ from CD45$^+$ lineage$^-$ lung cells and **i** the corresponding quantification presented as the absolute number of ILC2s. **j** Quantification of GATA-3 and **k** intranuclear Ki67 presented as MFI in WT and PD-1 KO aILC2s. Data are representative of at least two independent experiments and are presented as means ± SEM (*n* = 4; two-tailed Student's *t*-test or one-way ANOVA). Mouse image provided with permission from Servier Medical Art.

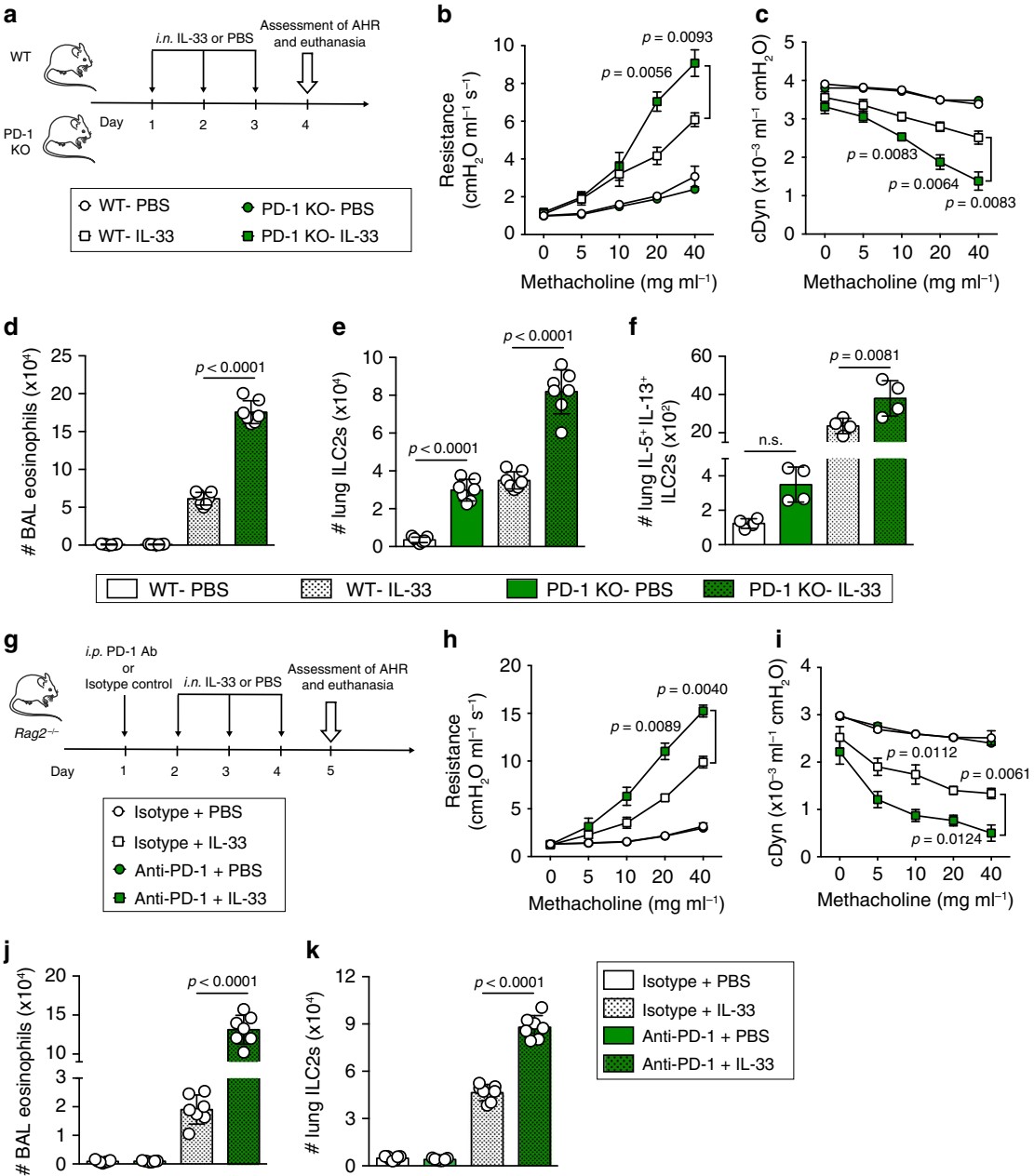

**Fig. 6 PD-1 expression on ILC2s ameliorates AHR and lung inflammation. a–f** WT and PD-1 KO mice were challenged intranasally for 3 consecutive days with 0.5 µg rm-IL-33. On day 4, AHR and lung inflammation were assessed. **b** Lung resistance and **c** dynamic compliance measured in restrained tracheostomized mechanically ventilated mice exposed to increasing concentrations of methacholine; $n = 4$. **d** Total number of eosinophils in BAL gated as CD45$^+$ SiglecF$^+$ CD11c$^-$; $n = 7$. **e** Total number of pulmonary ILC2s gated as lineage$^-$ CD45$^+$ ST2$^+$ CD127$^+$; $n = 7$. **f** Total number of IL-5$^+$ and IL-13$^+$ ILC2s identified by intracellular staining; $n = 4$. **g–k** Rag2$^{-/-}$ mice received intraperitoneal injection (i.p.) of anti-PD-1 blocking antibody (500 µg) or isotype control at day 1. Then mice were challenged intranasally on day 2–4 with 0.5 µg rm-IL-33. On day 5, AHR and lung inflammation were assessed. **h** Lung resistance and **i** dynamic compliance in response to increasing concentrations of methacholine; $n = 4$. **j** Total number of eosinophils in BAL and **k** total number of pulmonary ILC2s assessed by flow cytometry; $n = 7$. Data are representative of at least two independent experiments and are presented as means ± SEM (two-tailed Student's $t$ test or one-way ANOVA). Mouse image provided with permission from Servier Medical Art.

expression by a small population of nILC2s. Altogether, these results indicate that PD-1 controls IL-33-induced lung inflammation.

To exclude other lymphoid cells that highly express PD-1 and to address its specific role in ILC2 regulation, a cohort of Rag2$^{-/-}$ mice was treated with either a mouse PD-1 antagonist or corresponding isotype, and then challenged with IL-33 or PBS for 3 consecutive days. On day 5, AHR, BAL eosinophils, and pulmonary ILC2 numbers were measured (Fig. 6g). Consistent

with our previous observations with PD-1 KO mice, we demonstrated that anti-PD-1-treated Rag2$^{-/-}$ mice exhibited significantly greater lung resistance in response to increasing concentrations of methacholine, lower dynamic compliance, higher eosinophil recruitment, and higher lung ILC2 numbers as compared to control mice (Fig. 6h–k). The importance of the PD-1 axis in ILC2s was also confirmed in another asthma model induced by the fungal allergen *Alternaria alternata*. Briefly, this clinically relevant allergen strongly induced PD-1 expression on

ILC2s, and PD-1 blockade in $Rag2^{-/-}$ mice resulted in exacerbated AHR and lung inflammation (Supplementary Fig. 3).

**PD-1 agonist decreases ILC2-dependent AHR in humanized mice.** Since PD-1 plays a beneficial regulatory role in auto-immune diseases[39,40] and in our asthma model, we developed a human PD-1 agonist to target PD-1+ ILC2s in AHR. Human ILC2s were identified and sorted as CD45+ lineage− CD127+ CRTH2+ (Fig. 7a). We first observed that PD-1 was expressed on human ILC2s cultured in presence of recombinant human (rh)-IL-2 and rh-IL-7, and was importantly induced in response to rh-IL33 (Fig. 7b). It is important to mention here that freshly isolated ILC2s from healthy donors do not express PD-1 (Supplementary Fig. 4A). Second, we examined the effect of the PD-1 agonist on Th2 cytokine production by human ILC2s in vitro. Interestingly, IL-5 and IL-13 production was strongly suppressed in response to PD-1 agonist compared to the isotype (Fig. 7c, d). Lastly, we tested the in vivo efficiency of the PD-1 agonist in a humanized mouse model. Briefly, $Rag2^{-/-}$ $Il2rg^{-/-}$ mice that lack lymphoid cells were reconstituted with human peripheral blood ILC2s 1 day prior to receiving a PD-1 agonist or isotype injection for 4 consecutive days. Mice were also intranasally challenged during the last three days with PBS or rh-IL-33 to induce AHR (Fig. 7e). Of note, $Rag2^{-/-}$ $Il2rg^{-/-}$ mice do not develop AHR in the absence of any adoptive transfer (Supplementary Fig. 4B). As a validation of the humanized mouse model, intranasal administration of IL-33 significantly increased lung resistance and mouse eosinophil recruitment. Strikingly, lung resistance in PD-1 agonist-treated mice was significantly reduced as compared to isotype-treated mice (Fig. 7f). In line with lung resistance, dynamic compliance displayed higher response in PD-1 agonist-treated mice (Fig. 7g). Furthermore, agonist-treated mice showed abrogated lung inflammation associated with fewer number of human ILC2s, decreased eosinophilia, and decreased thickness of the airway epithelium (Fig. 7h–m). House dust mite (HDM) was also included in our study as a clinically relevant allergen, and the efficiency of PD-1 agonist was successfully confirmed in HDM-induced allergic asthma model, according to an established protocol[41]. $Rag2^{-/-}$ $Il2rg^{-/-}$ mice were reconstituted with total peripheral blood mononuclear cells (PBMCs) from HDM-allergic patients and intranasally challenged with HDM, as described in Supplementary Fig. 4C. Treatment of recipients with PD-1 agonist significantly reduced airway hyperreactivity and BAL eosinophils (Supplementary Fig. 4D, E). Moreover, we implemented the same treatment in a therapeutic protocol through the administration of PD-1 agonist one day after the last HDM challenge. This led to a significant amelioration of AHR and lung inflammation as well (Supplementary Fig. 4F–H). Collectively, these studies demonstrate for the first time that agonistic activation of PD-1 suppresses AHR in IL-33- and HDM-induced asthma models, and crucially suggest a new specific therapy for asthmatic allergies.

## Discussion
In this study, we characterized PD-1 implication in ILC2-dependent asthma, focusing on its capacity to regulate ILC2 activation and proliferation through metabolic balance. We have demonstrated that PD-1 regulates ILC2s to ameliorate AHR and control lung inflammation. Therefore, we developed a PD-1 agonist and demonstrated its efficiency as an ILC2 repressor in a humanized mouse model of asthma.

Recently, PD-1 was reported as an early checkpoint in ILC2 progenitors[16]. Here, we further characterized the PD-1 axis in pulmonary ILC2s. A small population of ILC2s expresses PD-1 at the steady state, remarkably further induced by IL-33;

therefore, PD-1 could serve as an activation marker and immune checkpoint in pulmonary ILC2s. In addition, this demonstrates that IL-33 is a potent PD-1 inducer in ILC2s, replacing the TCR antigen-mediated activation of PD-1 in effector T cells. We also carefully assessed PD-1 ligand expression in the lungs. IL-33 administration leads to a significant increase in the percentage of PD-L1 and PD-L2 expressing cells. Among the immune cells, Gr-1+ cells that mainly include recruited neutrophils[42], represent the major PD-L1+ PD-L2- population. Interestingly, CD11c+ populations represent the majority of PD-L1/PD-L2 double positive cells. This is in line with previous studies describing the inducibility of PD-L1 and PD-L2 expression on lung dendritic cells and macrophages in allergic airway inflammation[43,44]. Although lungs can provide a heterogeneous source of PD-1 ligands, PD-1 crosslinking in ILC2s is dictated by anatomical proximity. Recently, micro-anatomic niches around lung bronchi were identified as sites for colocalization and possible interaction between ILC2s and dendritic cells (DCs)[45]. This may suggest that DCs provide the major immune source of PD-L1/PD-L2 to ILC2s in the lungs.

The switch from fatty acid β-oxidation (FAO) and oxidative phosphorylation (OXPHOS) to aerobic glycolysis is a hallmark of T-cell activation, providing optimal energy for proliferation and cytokine production. Activated T cells also upregulate glutaminolysis, pentose phosphate pathway, and catabolism of branched-chain amino acids[30,46]. PD-1 has been described as a metabolic regulator in T cells, switching their metabolism from aerobic glycolysis to lipolysis and enhancing therefore the oxidative environment[47,48]. Unlike T cells, ILC2s exhibit context-dependent metabolism that primarily relies on FA metabolism. In particular, ILC2s display a high uptake of FA, as compared to regulatory T cells (Tregs), while the inhibition of FAO, but not glycolysis, alters ILC2 effector functions[32–34]. Here we demonstrate that PD-1 deficiency upregulates the expression of several glycolysis-dependent genes and enhances aerobic glycolysis in aILC2s. In addition, PD-1 deficiency enhances amino acid degradation including glutamine and methionine catabolism, fueling ILC2 activation and proliferation. It is worthwhile to mention that methionine degradation is the major source of methyl groups associated with RNA and histone methylation needed for T-cell proliferation[49]. In our study, the abnormal increase in ILC2 proliferative capacity was associated with altered metabolism shifting toward the dominance of glycolysis and amino acid metabolism. In parallel, the blockade of glucose metabolism decreases the proliferation of PD-1-deficient ILC2s to normal levels; however, it does not significantly affect WT ILC2s. This suggests that PD-1 is a potent metabolic checkpoint, defining the dominant ILC2 metabolic pathways for activation and proliferation during allergic asthma. Although the inhibitory effect of PD-1 on ILC2 proliferative potential has been demonstrated[17], our study is the first to suggest a direct metabolism-dependent inhibition of ILC2 proliferation by PD-1.

The expression of PD-1 on different immune populations has important implications. PD-1 deficiency in aILC2s affects more than 840 genes, notably apoptosis-related genes and Th2-associated transcription factors genes, including *Gata-3* and *Stat5*. Consequently, the lack of PD-1 is associated with an enhanced survival and an increased expression of GATA-3. Interestingly, genes encoding transcription factors that are classically known to regulate PD-1 expression in antigen-activated T cells, including NFATC1, forkhead box protein O1 (FOXO1), T-bet, B lymphocyte-induced maturation protein 1 (BLIMP1), and the serine–threonine kinase glycogen synthase kinase 3 (GSK3)[28] are not differentially expressed. This indicates that the regulation of PD-1 expression in ILC2s differs from T cells and requires further consideration.

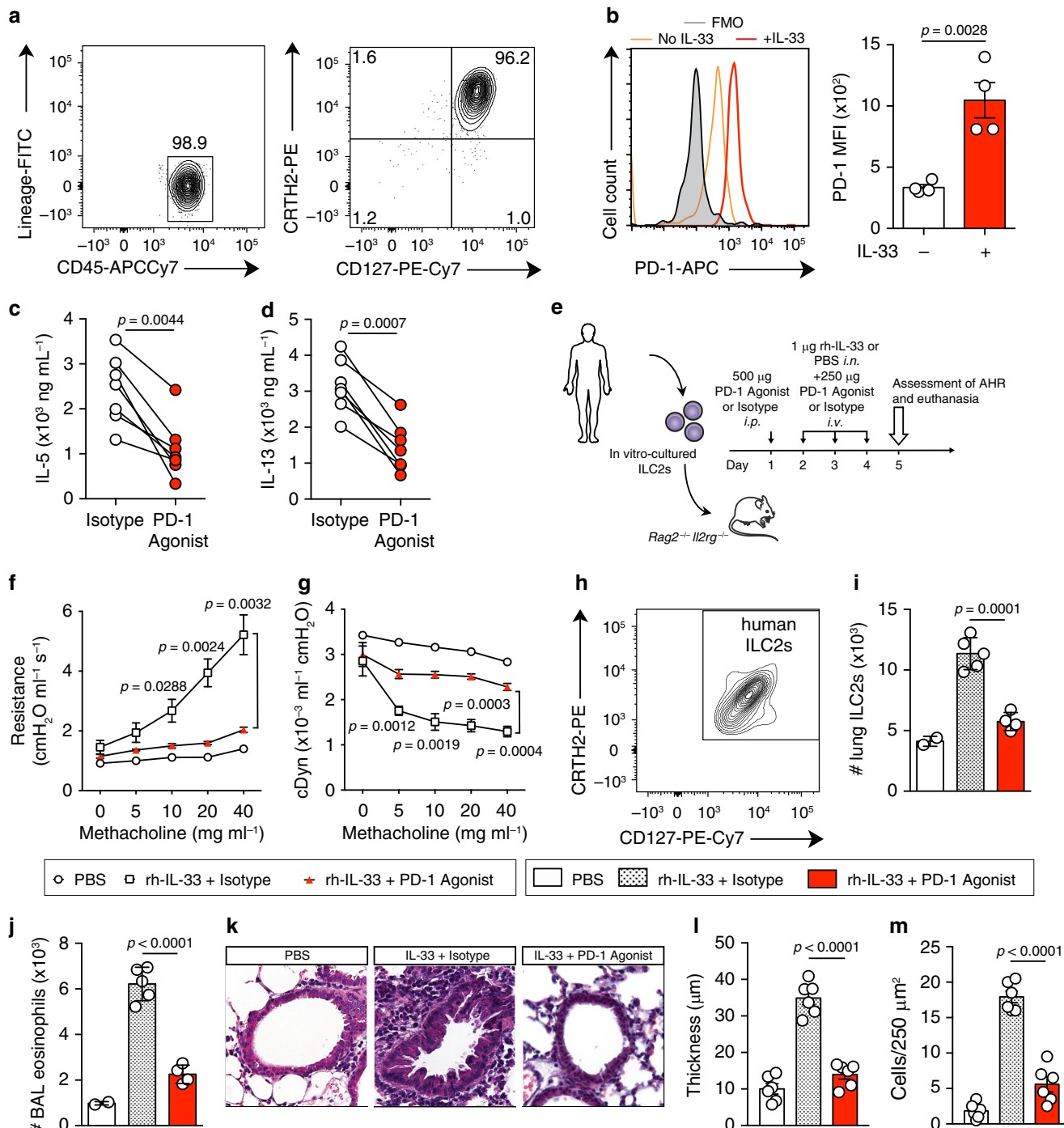

**Fig. 7 PD-1 is inducible on human ILC2s and PD-1 agonist suppresses ILC2-mediated AHR. a** Human ILC2s were FACS-sorted from PBMCs as CD45[+] lineage[−] CD127[+] CRTH2[+]. Purity was always ≥95%. **b−d** Cells were cultured ($5 \times 10^4$ per mL) in the presence of rh-IL-2 (10 ng mL[−1]) and rh-IL-7 (20 ng mL[−1]), with or without rh-IL-33 (20 ng mL[−1]), PD-1 agonist (25 μg mL[−1]), or corresponding isotype for 48 h. **b** Representative histogram of the expression of PD-1 in human ILC2s and corresponding quantification (right) presented as MFI; $n = 4$. **c** Levels of IL-5 and **d** IL-13 were quantified in culture supernatants from isotype- and PD-1 agonist-treated human ILC2s; $n = 7$. **e−m** In vitro cultured human ILC2s were adoptively transferred (intravenously) into *Rag2*[−/−] Il2rg[−/−] mice. At day 1, mice received an intraperitoneal injection of PD-1 agonist or control isotype (500μg). Then, mice were intranasally challenged with rh-IL-33 or PBS and injected with PD-1 agonist or isotype (250 μg) on days 2–4. Measurement of lung function and inflammation followed on day 5. **f** Lung resistance and **g** dynamic compliance measured in restrained tracheostomized mechanically ventilated mice exposed to increasing concentrations of methacholine; $n = 4$. **h, i** Total number of human ILC2s in lungs gated as CRTH2[+] CD127[+]; $n = 5$. **j** Total number of eosinophils in BAL gated as CD45[+] SiglecF[+] CD11c[−]; $n = 5$. **k** Lung histology (Scale bars, 50 mm); representative of two independent experiments. **l** Quantification of airway epithelium thickness and **m** infiltrating cells; $n = 6$. Data are representative of at least two independent experiments and are presented as means ± SEM (two-tailed Student's *t* test or one-way ANOVA). Mouse and human images provided with permission from Servier Medical Art.

ILC2s produce high levels of Th2 cytokines upon GATA-3 activation in response to pro-allergenic cytokines including IL-25 and IL-33[9]. PD-1 strongly dampens Th2 cytokine production in allergen-specific human T cells[40]. Our data demonstrate, at the transcriptional and the protein levels, that the high activation of GATA-3 in PD-1 deficient ILC2s is associated with a dramatic increase in Th2 cytokine production. This is consistent with other studies showing that the number of IL-5[+] and IL-13[+] ILC2s increases in the absence of PD-1 in IL-33-treated mice[17] and in obese mice[50]. In parallel, the engagement of PD-1 pathway using PD-L2 Fc strongly inhibits Th2 cytokine secretion, suggesting that the expression of PD-1 ligands in the lungs could significantly contribute to the regulation of ILC2 activation. Moreover, our study reveals that ILC2s can also provide a source of PD-L1, allowing for the autoregulation of ILC2 effector functions. Although a recent study has suggested *cis* interactions between PD-1 and PD-L1 on tumor cells and tumor-infiltrating antigen presenting cells[51], future studies are required to uncover and characterize the *cis* and *trans* interactions of PD-1 in pulmonary ILC2s. In particular, therapeutic strategies that could ensure an optimal engagement of PD-1 on ILC2s would directly reduce asthma pathogenesis, notably IL-5-mediated eosinophil recruitment and IL-13-dependent mucus metaplasia in the respiratory tract.

Few studies have documented the expression of PD-1 on T cells in asthmatic patients[52,53]. A negative correlation between PD-1 low expression on CD4[+] T cells and high specific IgE concentrations has been defined in human allergic asthma[54]. Our ex vivo results in mice show clearly that PD-1 axis strongly affects ILC2 activation and metabolism. This correlates with exacerbated AHR and lung inflammation in PD-1-deficient mice. In particular, ILC2 accumulation in lung was relatively high in naïve and IL-33-treated PD-1 deficient mice, illustrating the negative impact of PD-1 on ILC2 proliferation. Additionally, our $Rag2^{-/-}$ mouse model treated with PD-1 antagonist demonstrates that PD-1 regulates lung inflammation primarily through ILC2 inhibition, independently of B and T cells. Alternaria-induced asthma was also tested as a clinically relevant model to confirm the importance of the PD-1 axis in this inflammatory disease. Altogether, our data clearly reveal that PD-1 signaling leads to a less prominent Th2 profile in pulmonary ILC2s and suggest that a specific agonist of PD-1 could be a potential therapeutic strategy in ILC2-dependent asthma.

A recent study has suggested that the depletion of PD-1[+] cells using an anti-PD-1 coupled to immunotoxin helps controlling autoimmune diseases[55]. Alternatively, we hypothesized that the development of a PD-1 agonist has the potential to exploit PD-1 inhibitory signals to specifically inhibit highly activated immune cells. Our results robustly support an active and suppressive role of PD-1 in airway inflammation, spurred development of a specific agonist for human PD-1 in asthma. To validate this tool, we used a humanized mouse model that gives a reliable prediction of human responses in clinical development. Our findings in the humanized mouse model demonstrate that PD-1 agonist elicits an efficient suppression of activated ILC2s and suppresses AHR as well as lung inflammation in IL-33 and HDM-induced asthma. Strikingly, PD-1 agonist is efficient in ameliorating AHR in the context of both preventive and therapeutic protocols. Therefore, our clinically relevant approaches support the potential clinical use of PD-1 agonist for asthma treatment.

In conclusion, this study highlights the importance of PD-1 axis in pulmonary ILC2s, demonstrating that PD-1 counters ILC2 activation and functions through different mechanisms, including metabolic modulation. Therefore, PD-1 plays a critical regulatory role in AHR, mainly through ILC2 inhibition. The success of a PD-1 agonist in humanized mouse model of allergic asthma underscores the important role of PD-1 in AHR and highlights the therapeutic potential of PD-1 agonists for the treatment of allergic asthma. This strategy may also provide a new opportunity to achieve an overarching clinical goal in allergies and autoimmunity.

## Methods

**Mice**. Wild-type (WT) BALB/cByJ, *Rag2*-deficient ($Rag2^{-/-}$), and *Rag2/Il2rg* double knockout ($Rag2^{-/-}$ $Il2rg^{-/-}$) mice on the BALB/c background mice were purchased from Jackson Laboratory (Bar Harbor, ME). PD-1-knockout (KO) BALB/c mice were generated in the Sharpe laboratory[56]. All mice were bred in the animal facility of the Keck School of Medicine, University of Southern California (USC). Mice were maintained at macroenvironmental temperature of 21–22 °C, humidity (48–52%), in a conventional 12:12 light/dark cycle with lights on at 6:00 a.m. and off at 6:00 p.m. Four to eight-week-old aged and sexed-matched mice were used in the study. All experimentation protocols were approved by the USC Institutional Animal Care and Use Committee and conducted in accordance to the principles of the Declaration of Helsinki.

**Induction and assessment of AHR**. Mice were sensitized intranasally for 3 days with carrier-free recombinant mouse (rm)-IL-33 (0.5 μg per mouse in 50 μL, BioLegend) to induce AHR or with PBS, as the control. In some experiments, mice were challenged for 4 consecutive days with 100 μg of *Alternaria alternata* extracts (Greer Laboratories). To block glycolysis in vivo, mice were injected intraperitoneally on 2 consecutive days with 500 mg kg$^{-1}$ of 2-Deoxyglucose (2-DG) (Sigma-Aldrich) or with PBS (vehicle). For the neutralization models, $Rag2^{-/-}$ mice received an intraperitoneal injection of PD-1 blocking antibody (500 μg per mouse, clone 29 F.1A12; BioXcell) or a rat IgG2aκ isotype control 1 day before rm-IL-33 or Alternaria intranasal administration.

On day 4, lung function was evaluated by direct measurement of lung resistance and dynamic compliance (cDyn) in restrained, tracheostomized, and mechanically ventilated mice using the FinePointe RC system (Buxco Research Systems) under general anesthesia. Mice were sequentially challenged with aerosolized PBS (baseline), followed by increasing doses of methacholine ranging from 5 to 40 mg mL$^{-1}$. Maximum lung resistance and minimum compliance values were recorded during a 3-min period after each methacholine challenge. AHR data were analyzed by repeated measurements of a general linear model.

**BAL collection and lung preparation for flow cytometry**. After measurements of AHR, lungs were injected with 3 ml ice-cold PBS to collect BAL cells. The following antibodies were used to identify eosinophils: PE-Cy7 CD45 (clone 30-F11; Bio-Legend), APC-Cy7 CD11c (clone N418; BioLegend) and PE Siglec-F (clone E50-2440; BD Biosciences). Lung tissue was cut into small pieces and incubated in type IV collagenase (1.6 mg ml$^{-1}$; Worthington Biochemicals) at 37 °C for 60 min. Single-cell suspensions were obtained by passing the lung tissue digest through a 70-μm cell strainer. The following antibodies were used to identify ILC2s: biotinylated anti-mouse TCR-γδ (clone eBioGL3; eBioscience), TCR-β (clone H57-597), CD3e (clone 145−2C11), CD45R (clone RA3-6B2), Gr-1 (clone RB6-8C5), CD11c (clone N418), CD11b (clone M1/70), Ter119 (clone TER-119), FcεRI (clone MAR-1), CD5 (clone 53-7.3), NK1.1 (clone PK136), Streptavidin-FITC, PE-Cy7 or APC anti-mouse CD127 (clone A7R34), APC-Cy7 anti-mouse CD45 (clone 30-F11) (all from BioLegend), and PerCP-eFluor710 anti-mouse ST2 (clone RMST2-2; eBioscience). Fluorescent live/dead fixable stains (ThermoFisher Scientific) were used to exclude dead cells, according to manufacturer's instructions. Brilliant Violet (BV) 421 (clone J43; BD Biosciences) or PE anti-PD-1 (clone J43; eBioscience), APC or PE-Cy7 anti-mouse PD-L1 (clone 10 F.9G2; BioLegend), PE anti-mouse PD-L2 (clone TY25; BioLegend), BV421 anti-mouse CD11b (clone M1/70; BD Biosciences), PerCP-Cy5.5 anti-mouse CD11c (clone N418; BioLegend) and APC anti-mouse Gr-1 (clone RB6-8C5; BioLegend) were used in some experiments. Intranuclear staining with APC anti-mouse Ki67 (clone SolA15, eBioscience) and PE anti-mouse GATA-3 (clone TWAJ; Invitrogen) was performed using the Foxp3 Transcription Factor Staining Kit (ThermoFisher Scientific). For apoptosis, cells were stained with AnnexinV-PE (Invitrogen) and DAPI (Sigma). The BD Cytofix/Cytoperm Kit (BD Biosciences) was used according to the manufacturer instructions to perform intracellular staining with PE-Cy7 anti-Bcl2 (clone 10C4; eBioscience). Intracellular staining was also performed to assess cytokine production using PE anti-mouse/human IL-5 (clone TRFK5; BioLegend) and eFluor450 anti-mouse IL-13 (clone eBio13A; eBioscience) after 4 h of in vitro incubation with 50 μg mL$^{-1}$ PMA (Sigma), 500 μg mL$^{-1}$ ionomycin (Sigma), and 1 μg mL$^{-1}$ Golgi plug. CountBright Absolute Count Beads were used to count lung immune cells (Invitrogen). Antibodies were added at a 1:200 dilution, when no recommendations were provided by the manufacturer. Acquisition was performed on a BD FACS-Canto II (BD Biosciences) using the BD FACSDiva software v8.0.1. Data were analyzed with FlowJo software (TreeStar) version 10.

**ILC2 sorting and in vitro culture**. Murine ILC2s were sorted on a FACSARIA III system (Becton Dickinson) based on the lack of expression of classical lineage markers (CD3e, CD45R, Gr-1, CD11c, CD11b, Ter119, CD5, TCR-β, TCR-γδ,

NK1.1, and FcεRI) and expression of CD45, ST2, and CD127. Sorted ILC2s were cultured at 37 °C, in the presence of rm-IL-2 (10 ng mL⁻¹) and rm-IL-7 (10 ng mL⁻¹) in RPMI (Gibco) supplemented with 10% heat-inactivated fetal bovine serum (FBS) and 100 units per mL penicillin–streptomycin. Naïve ILC2s were stimulated in vitro for 48 h with 20 ng mL⁻¹ of rm-IL-33 and incubated with or without PD-1 blocking antibody (10 μg mL⁻¹, clone 29 F.1A12; BioXcell) or a rat IgG2aκ isotype control. In some experiments, 0.5 mM of 2-DG or 5 mM of cycloleucine (CYL) were added to the culture for 24 h.

**RNA-sequencing and data analysis**. Freshly sorted ILC2s after three intranasal administrations of rm-IL-33 (0.5 μg per mouse), defined in this study as activated ILC2s (aILC2s), were incubated (5 × 10⁴ per mL) with rm-IL-2 (10 ng mL⁻¹) and rm-IL-7 (10 ng mL⁻¹) for 24 h. Total RNA was isolated using MicroRNAeasy (Qiagen). In total, 10 ng of input RNA was used to produce cDNA for downstream library preparation. Samples were sequenced on a NextSeq 500 (Illumina) system. Raw reads were aligned, normalized and further analyzed using Partek Genomics Suite software, version 7.0 Copyright; Partek Inc. Normalized read counts were tested for differential expression using Partek's gene-specific analysis (GSA) algorithm.

**Supernatant cytokine measurement**. aILC2s were cultured (5 × 10⁴ per mL) for 24 h in the presence of rm-IL-2 (10 ng mL⁻¹) and rm-IL-7 (10 ng mL⁻¹). When indicated, ILC2s were incubated in PD-L2 Fc or isotype-coated wells (5 μg mL⁻¹, R&D systems). The levels of IL-5, IL-9, and IL-13 were measured in supernatants using Legendplex multiplex kits (BioLegend) and data were analyzed via the LEGENDplex data analysis software v8.0. Human ILC2s were sorted and cultured for 48 h with rh-IL-2 (20 ng mL⁻¹), rh-IL-7 (20 ng mL⁻¹), rh-IL-33 (20 ng mL⁻¹), and PD-1 agonist or control isotype (10 μg mL⁻¹). Human IL-5 and IL-13 were quantified in supernatants by ELISA (ThermoFisher Scientific). The PD-1 agonist has been fully characterized[57]. Briefly, the antibody binds to human PD-1 with an equilibrium dissociation constant (KD) of $5 \times 10^{-8}$ M; an association constant (ka) of $\sim 3 \times 10^4$ l Ms⁻¹ and a dissociation constant (kd) of $\sim 3 \times 10^{-3}$ l s⁻¹. To obtain antibody please contact Janssen Pharmaceuticals.

**Glucose uptake assay and measurement of glycolytic functions**. aILC2s were incubated (5 × 10⁴ per mL) with rm-IL-2 (10 ng mL⁻¹) and rm-IL-7 (10 ng mL⁻¹) for 24 h. To assess glucose uptake, 50 μg mL⁻¹ of 2-(N-[7-nitrobenz-2-oxa-1,3-diazol-4-yl] amino)−2-deoxyglucose (2-NBDG) from ThermoFisher Scientific were added to the culture for 20 min. Acquisition was performed using a BD FACSCanto II (BD Biosciences). The expression of the glucose transporter 1 (Glut-1) was assessed using a PE-Cy7 anti-Glut-1 (polyclonal, Novus Biologicals). The real-time extracellular acidification rate (ECAR) and oxygen consumption rate (OCR) were measured using a XF 96 extracellular flux analyzer (Seahorse Bioscience). Briefly, 5 × 10⁴ to 15 × 10⁴ aILC2s were plated in Seahorse media supplemented with 1 mM pyruvate, 2 mM glutamine, and 10 mM glucose. Mito stress test kit (Agilent, San Diego CA), using 1 μM oligomycin, 1 μM FCCP, 1 μM etomoxir, and 0.5 μM rotenone/antimycinA, was performed according to the manufacturer's instructions.

**LC-MS metabolomics**. Freshly sorted aILC2s were incubated with rm-IL-2 (10 ng mL⁻¹) and rm-IL-7 (10 ng mL⁻¹) for 24 h. Metabolites were extracted from 1 × 10⁶ isolated ILC2s by addition of 500 μL of ice-cold 80% aqueous methanol. Targeted metabolomics of polar, primary metabolites was performed on a TQ-XS triple quadrupole mass spectrometer (Waters) coupled to an I-class UPLC system (Waters). Separations were carried out on a ZIC-pHILIC column (2.1 × 150 mm, 5 μM; EMD Millipore). The mobile phases were (A) water with 15 mM ammonium bicarbonate adjusted to pH 9.6 with ammonium hydroxide and (B) acetonitrile. The flow rate was 200 μL min⁻¹ and the column was held at 50 °C. The injection volume was 1 μL. The gradient was as follows: 0 min, 90% B; 1.5 min, 90% B; 16 min, 20% B; 18 min, 20% B; 20 min, 90% B; and 28 min, 90% B. The MS was operated in selected reaction monitoring mode (SRM). Source and desolvation temperatures were 150 °C and 600 °C, respectively. Desolvation gas was set to 1100 L h⁻¹ and cone gas to 150 L h⁻¹. Collision gas was set to 0.15 mL min⁻¹. All gases were nitrogen except the collision gas, which was argon. Capillary voltage was 1 kV in positive ion mode and 2 kV in negative ion mode. A quality control sample, generated by pooling equal aliquots of each sample extract, was analyzed every 3–4 injections to monitor system stability and performance. Samples were analyzed in random order. Data processing (peak integration) was performed with Skyline software. When available, two SRM transitions were monitored for each metabolite and the peak areas were summed.

**Human ILC2 isolation and adoptive transfer**. All human studies were approved by USC Institutional review board and conducted in accordance to the principles of the Declaration of Helsinki. Informed consent was obtained from all human participants. Human peripheral blood ILC2s were isolated from total PBMCs as described previously[11]. Briefly, human fresh blood was first diluted 1:1 in PBS and transferred to SepMateTM-50 separation tubes (STEMCELL Technologies)

prefilled with 15 mL Lymphoprep™ (Axis-Shield). Samples were centrifuged at 1200 × g for 10 min to collect PBMCs. The CRTH2 MicroBead kit (Miltenyi Biotec) was then used according to the manufacturer's conditions in order to isolate CRTH2⁺cells. Human ILC2s were sorted from CRTH2⁺ cells based on the lack of expression of classical lineage markers and expression of CD45, CRTH2 and CD127. The following antibodies were used (all from BioLegend): FITC anti-lineage, FITC anti-CD235a (clone HI264), FITC anti-FceRIa (clone AER-37), FITC anti-CD1a (clone HI149), FITC anti-CD123 (clone 6H6), APC-Cy7 anti-CD45 (clone HI30), PE anti-CD294 (CRTH2) (clone BM16), PE-Cy7 anti-CD127 (clone A019D5), and APC anti-CD279 (PD-1) (clone EH12.2H7). Purified human ILC2s were cultured with rh-IL-2 (20 ng mL⁻¹) and rh-IL-7 (20 ng mL⁻¹) for 3 days then adoptively transferred to $Rag2^{-/-} Il2rg^{-/-}$ mice (5 × 10⁴ cells per mouse). On day 1, mice received either human PD-1 agonist (500 μg per mouse, Janssen Pharmaceuticals) or isotype-matched control then 250 μg per mouse for 3 consecutive days (2–4) along with intranasal administration of 1 μg rh-IL-33. On day 5, lung function was measured before BAL and lungs were collected for analysis. In some experiments, $Rag2^{-/-} Il2rg^{-/-}$ mice were reconstituted with total PBMCs from HDM-allergic patients. Briefly, 5 × 10⁶ PBMCs were stimulated in vitro with 100 μg mL⁻¹ of HDM extracts (Greer Laboratories) for 3 days prior to injection in mice[41,58]. At days 3 and 4, reconstituted mice were intranasally sensitized with 200 μg of HDM extracts then injected with human PD-1 agonist (250 μg per mouse, Janssen Pharmaceuticals) or isotype-matched control. At days 9, 10 and 11, mice were challenged with 100 μg of HDM extracts before AHR and lung inflammation assessment at day 12. For the therapeutic protocol, HDM-challenged mice were intraperitoneally treated with human PD-1 agonist or isotype-matched control (250 μg per mouse) at day 12, before AHR and lung inflammation assessment at day 14.

**Histology**. When indicated, one lobe per lung was collected for histology and stored in paraformaldehyde 4% buffered in PBS. Lungs were embedded in paraffin and cut into 4-μm sections for hematoxylin and eosin (H&E) staining. Histology pictures were acquired on a Leica DME microscope and Leica ICC50HD camera (Leica) and ImageJ was used for analysis.

**Statistical analysis**. A two-tailed Student's $t$ test for unpaired data was applied for comparisons between 2 groups. For multigroup comparisons, we used one-way ANOVA with the Tukey post hoc test. Data were analyzed with Prism Software (GraphPad Software Inc.). Error bars represent standard error of the mean. $p$ value < 0.05 was considered to denote statistical significance.

**Reporting summary**. Further information on research design is available in the Nature Research Reporting Summary linked to this article.

## Data availability
RNA sequence data that support the findings of this study have been deposited in Genbank with the primary accession code GSE153909. All other data are present in the article and its Supplementary files, or are available from the corresponding author upon reasonable request. Source data are provided with this paper.

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

## Acknowledgements

This article was financially supported by National Institutes of Health Public Health Service grants R01 ES025786, R01 ES021801, R01 HL144790, and R21 AI109059 (O.A.). We would like to thank Dr. Ling-Yang Hao for reviewing the manuscript. We are grateful to USC Libraries Bioinformatics Service for assisting with data analysis, in particular Dr. Yibu Chen. The bioinformatics software and computing resources used in the analysis are funded by the USC Office of Research and the Norris Medical Library. We thank Dr. Jay Kirkwood from the metabolomics core facility at the University of California Riverside for the technical support.

## Author contributions

D.G.H. designed and performed experiments, analyzed results and wrote the manuscript. R.L. and P.S.J. equally contributed by performing experiments and analyzing data. B.P.H., E.H., L.G.T. and J.D.P. contributed to interpretation of the data and provided animal husbandry for experiments. P.S. and G.L. contributed to interpretation of data and critically reviewed the manuscript. A.H.S. provided the PD-1 KO mice and critically reviewed the article. O.A. supervised, designed the experiments, interpreted the data, and critically reviewed the manuscript. All authors contributed to manuscript revision.

## Competing interests

A.H.S. declares that they have patents/pending royalties on the PD-1 pathway from Roche and Novartis. G.L. and P.S. declare that they are employees of Janssen R&D. O.A. declares that they receive grant support from the NIH and Janssen Pharmaceuticals. The other authors declare no competing interests.
