## [Peer Review File · Nature Communications]

Reviewers' comments:

Reviewer #1 (Remarks to the Author):

In the manuscript titled "PD-1 pathway regulates ILC2 metabolism and PD-1 agonist treatment ameliorates airway hyperreactivity" the authors investigate the previously described role of the PD-1/PD-L1 axis in negatively regulating allergic airway disease pathogenesis. Specifically, they demonstrate that ILC2 PD-1 engagement is necessary to prevent a shift in metabolism towards glycolysis (as has been shown in T cells), which was shown to be required for both expression of the master transcriptional regulator GATA3 and proliferation. Further, the authors perform in vivo studies demonstrating the utility of a novel PD-1 agonist in the treatment of allergic airway disease. Importantly, this therapeutic approach was tested with humanized mice and nearly abrogated airway hyperresponsiveness and inflammation, and thus represents a candidate pathway for future drug development. However, addressing several points below would significantly improve the impact of the work.

Major comments:

- 1) Lack of clinically relevant allergen. The authors utilize IL-33 throughout the study in order to model allergic airway disease pathogenesis. While the IL-33-induced response can be similar to a response-induced by a clinically relevant allergen, this is not necessarily the case. Thus, the authors should perform studies demonstrating the importance of the PD-1/PD-L1 axis with a clinically relevant allergen, such as HDM or Alternaria. These studies would also provide insight into how PD-1 might regulate epithelial cytokines.
- 2) Lack of mechanism for PD-1 regulation of ILC2 responses through metabolic changes in vivo. Though the association with metabolic changes in ILC2s regulated by PD-1 are interesting, it is not clear that PD-1 regulation of ILC2 airway responses and the asthma phenotype in vivo occurs through these ILC2 metabolic changes.
- 3) Unknown effect of PD-1 agonism on non-ILC2s. The authors demonstrate that the PD-1 agonist nearly abrogates allergic airway disease pathogenesis in a humanized mouse model. However, given that the agonist only targets human PD-1 and that the only human PD-1 in this system is expressed by the human ILC2s, how would mouse PD-1 agonism perform in vivo in a non-humanized model? Would disease still be abrogated? This is important to understand if we are to administer PD-1 agonists in humans which have other PD-1 expressing cells in addition to ILC2s. Additionally, histological staining for inflammation and mucous hypersecretion would help readers assess the utility of PD-1 agonism in the treatment of disease.
- 4) Unknown endogenous PD-L1 signal. The authors demonstrate the necessity and sufficiency of the PD-1/PD-L1 axis in dampening allergic airway inflammation; however, they do not show which cells are the physiologic source of PD-L1 in the asthma model.

Minor comments:

- 1) A few confusing sentences and grammatical errors. I mention a couple comments regarding this issue in my below minor comments, however, I suggest that the authors carefully proofread and edit the entire manuscript again in order to make the manuscript as easy to follow as possible.
- 2) Line 115: Suggest changing to "...suggesting a decrease in controlled...".
- 3) Line 145: Suggest removing "did".
- 4) Line 154: Suggest changing "improved" to "altered".
- 5) Line 165: Suggest changing "utilization" to "metabolism".
- 6) Line 188: Suggest changing "important" to something more appropriate.
- 7) Line 245/341: Suggest changing "proved" to something more appropriate.

- 8) Line 260: Suggest changing "sensitized" to "challenged".
- 9) Line 274-275: Suggest rephrasing.
- 10) Line 284-88: These sentences do not appear to be clearly related to the work's findings.
- 11) Line 317: Do the authors mean to say these are not differentially expressed? Please clarify.
- 12) Line 319: This is a bit confusing as ILC2 PD-1 expression is also context dependent (e.g. IL-33 induced). Suggest removing "in which the involvement of PD-1 in Th1 and Th2 polarization is context dependent".
- 13) Line 340: Suggest changing to independent of B/T cells.
- 14) Line 665: Suggest removing 'in'

Reviewer #2 (Remarks to the Author):

Using mouse models, this manuscript by Helou et al reports two main relevant findings: (1) The inhibitory immune receptor PD-1 restrains the proliferation and effector function of group 2 innate lymphoid cells (ILC2s) in allergic airway inflammation by acting on ILC2 metabolism. (2) PD-1 agonism is tested as a potential new way to treat asthma. The study is based on well-designed experiments and the central conclusions are supported by the data. While the finding that PD-1 inhibits ILC2 function is not novel, the authors provide a new mechanism of how this occurs, i.e. through preventing the metabolic reprogramming that is necessary for ILC growth and cytokine production. Investigation of ILC metabolism is a topical area and the findings reported in the manuscript are of interest to the readership of Nature Communications. To further strengthen the study, the following comments should be addressed.

Major comments

- (1) Figure 1 and Figure 4 show that PD-1 limits the survival and proliferation of activated ILC2s, respectively. It is unclear whether the enhanced proliferation of PD-1 knockout ILC2s is due to better survival or due to a selective effect of PD-1 on proliferation. Cell death and proliferation should therefore be measured simultaneously in wild-type and PD-1-deficient ILC2s.
- (2) The authors show that activated ILC2s express both PD-1 and its ligand PD-L1 (Figure 1). Does PD-1 regulate ILC2 function in cis (through PD-L1 on ILC2s) or in trans (PD-1 ligands provided by other cells)?
- (3) Related to Figure 5: Does PD-1 play an ILC2-intrinsic role in regulating IL-33-induced allergic airway inflammation? The authors exclude TH2 (and B) cells by performing experiments in Rag2 knockout mice, but other PD-1-expressing cells, such as alveolar macrophages, could be involved.
- (4) Figure 5E shows that, in steady-state, the number of lung ILC2s is increased in PD-1 knockout mice. However, the authors report that naïve lung ILC2s hardly express PD-1 surface protein (Figure 1B). The authors should reconcile these observations. Taylor et al. J Exp Med 2017 published that lung ILC2s are increased in mice lacking PD-1, but also demonstrate that ~10% of lung ILC2s express surface PD-1 in steady-state.
- (5) In contrast to mouse lung ILC2s (Figure 1B), human blood ILC2s express PD-1 constitutively, i.e. in the absence of IL-33 treatment (Figure 6B). Is this a species-specific difference or is PD-1 induced by IL-2 and IL-7 that were used to culture human ILC2s in vitro? This is a relevant point since, in humans, PD-1 agonism might have unwanted effects on ILC2s in the absence of airway inflammation.
- (6) Related to Figure 6: Is PD-1 agonist treatment also effective in ameliorating ongoing allergic airway inflammation? This is relevant for possible asthma treatment in humans. To address this question, mice should be treated with PD-1 agonist at later time, i.e. after IL-33-induced airway

inflammation.

(7) Does IL-33 induce airway hyper-reactivity in Rag2 Il2rg double knockout mice in the absence of transferred human ILC2s (Figure 6)? In other words, is airway hyper-reactivity driven by the transferred human ILC2s or by residual IL-33-responsive mouse cells in Rag2 Il2rg double knockout mice?

Minor comments

(1) ILCs and T cells are related and many concepts established for T cells apply to ILC biology. However, ILCs also possess some unique features. Therefore, the similarities as well as potential differences between ILC2 and Th2 metabolism should be discussed.

(2) Statistics: ANOVA (not Student's t-test) should be used for multi-group comparisons in Figures 4E-F, 5D-F, 5J-K, and 6H.

(3) It is unclear why isotype-treated wild-type mice are used as controls for anti-PD-1-treated Rag2 knockout mice in Figure 5G-K. It seems more appropriate to use isotype-treated Rag2 knockout mice instead.

(4) Related to Figure 6H: Representative flow cytometry plots of human lung ILC2s should be shown in control and PD-1 agonist-treated mice.

(5) The legend of Figure 2 states n=6, whereas only 3-4 data points are shown in the figure.

(6) The legend for Figure 4 panels K and L is missing.

(7) In the methods section, information about antibody clones and concentrations should be added to facilitate the ability of other researchers to reproduce the work.

Reviewer #3 (Remarks to the Author):

The manuscript by Helou et al describes the role of PD-1 in regulating ILC-2 metabolism in airway hyperreactivity. The authors demonstrate that PD-1 signaling can block essential proliferative signalling pathways such as STAT5, apoptotic pathways and can modulate the metabolic profile of ILCs. The authors then explore the possibility that the increased proliferative potential of PD1 deficient ILC2s is due to differential metabolic capacity and show that blocking glucose (2DG) or methionine metabolism (cycloleucine) directly correlated to regulating ILC-2 proliferation.

The study employs significant amount of transcriptomics and metabolomics to define the mechanistic role of PD-1 in ILC-2 biology. It confirms previous observations that PD-1 (1) can inhibit ILC2 proliferation and (2) can inhibit ILC2 cytokine production (ref. 38).

The authors further define two new function for PD-1 in ILC-2s through transcriptomics data which include (a) regulation of apoptosis and (b) regulation of glycolysis.

The most notable and novel finding within this work is the use of PD-1 agonist as a therapeutic target for treating human ILC2 mediated AHR. There are several questions pertaining to the study which requires further clarifications.

While the role of PD-1 in ILC-2s have been previously shown, here the novelty clearly lies in the fact that PD-1 can regulate an apoptotic signature in ILC2s. Although the Annexin data is a clear functional test, whether the anti-apoptotic phenotype is a by-product of increased STAT5 signaling in response to IL2/IL7 is not clear. For instance, can the authors clarify if the anti-apoptotic

phenotype seen can be uncoupled from the STAT5 differences seen in transcriptomics data presented here.

The second interesting observation is the metabolic regulation of ILC-2s. However, the functional sea horse data does not provide a clear and in-depth analysis of the glycolytic capacity of these cells, thereby making it difficult to prove that PD-1 has an effect on ILC-2 metabolism.

The third major observation within the manuscript is the use of PD-1 agonists in AHR. A similar experimental murine model was performed in the Yu et al, Nature, 2016, Vol.539, 102-106; where the authors showed the opposite effect with papain challenge and influenza challenge. In both these models, it was shown that PD-1 blocking antibodies decreased ILC2 cytokine production leading to less eosinophil accumulation. Can the authors discuss the discrepancies between the two studies?

Specific Concerns

Figure 1.

The characterization of ILC2s as naïve, mature and inflammatory needs to be clearly defined. Was a live/dead gate utilised in the analysis and if yes it would be good to add the gating strategy. In plots H&I, the data shown are from n=4 mice, but the legend describes this as a cumulative data of 3 experiments. Is this representative or cumulative data? While it is hard to perform western blotting with such limited cell numbers, would it be possible to confirm any of the transcriptomics data with protein data for genes shown in Fig.1F?

Figure 2

The authors demonstrate that PD-1 regulates glycolysis but the data shown in Fig. I does not optimally demonstrate this phenomenon. The sea horse analysis simply shows mitochondrial respiration. It would be more appropriate to measure ECAR with glucose, oligomycin and 2DG? Further, the OCR/ECAR ratio is at maximal respiration (assuming after FCCP injection), what is the ratio for basal respiration? Identifying basal glycolytic rate, maximal glycolytic rate and glycolytic capacity may provide more insight into the role of PD-1 in glycolysis? Also, can the authors clarify what the y axis scale denotes in panels c-f? Is it fold change?

Figure 3

Please denote what the values on Y axis denotes in c-l.

Figure 4

Can the authors clarify the use of PDL2 FC over PDL1 FC?

Minor Comments

The introductory paragraph does not fully acknowledge some of the original literature on PD-1 signaling. For instance, the work by Okazaki et al and Chemnitz et al are one of the first to demonstrate that PD-1 cytoplasmic tail recruits SHP2. Recently, further work by Ronald Vale and Rafi Ahmed's group have clarified this signalling cascade. The PD-1 stop signal work was first demonstrated by Fife et al. Also, ref. 15, 16 are not the first observations on PD-1 expression on ILC-2s. These were first demonstrated by refs. 33, 38 and Seillet et al, Cell reports, 2016, 17 (2):436-447.

Reviewer #4 (Remarks to the Author):

Review for Helou, Akbari et al.:

Summary: The authors identify the checkpoint molecule PD-1 as an IL-33 driven activation marker of lung type 2 innate lymphoid cells (ILC2s) that negatively regulates ILC2 proliferation, survival, and production of type 2 cytokines within the lung. The authors show that PD-1 is upregulated in IL-33 activated lung ILC2s. In PD-1 deficient mice, ILC2s can proliferate, survive, and produce more cytokines in response to IL-33, independent of T and B cells. The authors demonstrate that a lack of PD-1 induces/heightens a switch in ILC2s to aerobic glycolysis and amino acid catabolism. The authors use a humanized mouse model to demonstrate that PD-1 agonism is sufficient to limit human ILC2s and allergic airway response in vivo.

Together, the work is well-written and the data are straightforward and convincing. This study is the first to suggest a metabolic mechanism behind PD-1 regulation in pulmonary ILC2s, potentially highlighting PD-1 agonism as a therapeutic potential for the control of allergic asthma. One drawback is the prior publications that cover similar concepts (ILC2s and PD-1 signaling). For example, Taylor, JEM 2017, which shows PD-1 KO ILC2s have enhanced STAT5 signaling, increased proliferation and type 2 cytokine production, and offer superior protection against helminth infection. Yu, Liu, Nature, 2016 found PD-1 was highly expressed on ILC progenitors, but also show that activated lung ILC2s express high levels of PD-1. Moral, Nature, 2020 recently found that ILC2s in pancreatic cancer express PD-1 in an IL-33 dependent manner, restricting ILC2 function and limiting anti-tumor immunity. Together, these published works demonstrate that IL-33 can activate ILC2s to upregulate PD-1, that PD-1 signaling limits ILC2 proliferation and cytokine production, and that loss of PD-1 in ILC2s (or PD-1 blockade) leads to increased ILC2 functionality/activity. Several elements of this current work overlap with these ideas, but there are also very interesting and novel points, including (1) the metabolic impact of PD-1 loss on ILC2s and (2) the potential ability to agonize PD-1 to restrict ILC2 function in the treatment of allergic asthma. Major and minor comments are listed below:

Major comments:

1) The metabolic analysis and metabolomics are clearly a strength of this work. However, aspects of the results are difficult to definitively interpret and put into context. How do these metabolic findings (Figures 2-4) compare to that of naïve, non-IL-33 treated ILC2s? How specific are these metabolic changes for lung ILC2s, for example in comparison with activated and differentiated Th2 cells? Are these ILC2 changes unique to PD-1 inhibition, or does this PD-1 KO ILC2 profile more generally reflect the differences between a more activated and less activated lung ILC2? For example, it seems plausible that more highly activated lung ILC2s (PD-1 KO ILC2s after IL-33 treatment) would be more dependent on glycolysis and amino acid catabolism, similar to other highly activated lymphocytes. These questions are meant to illustrate some of the conceptual points that could strengthen the author's conclusions and add to the novelty of the work.

2) The authors use their results with a PD-1 agonist in humanized mice to infer possible pathways within asthma patients. However, their conclusions would be strengthened by comparing their results with a better-established model of allergic asthma (e.g. House dust mite, papain, fungal infection, chitin etc) to determine the impact of PD-1 signaling. Although IL-33 administration is a clean system, its physiologic relevance is less clear.

3) In Figure 5D, there seems to be a difference in number and cytokine expression between naïve PD-1KO and WT ILC2s, suggesting that PD-1 has impacts on ILC2s (or other cells that indirectly impact ILC2s) at steady state. Therefore, a complicating factor in the difference between IL-33 treated WT and PD-1KO ILC2s is their baseline activation state. To address these concerns, the authors could provide further characterization of PD-1KO mice at steady state. In a related point, the contribution of ILC2-intrinsic PD-1 signaling, versus signaling on other cellular targets, is not well-established in this work. Some effort to determine direct (ILC2-intrinsic) versus indirect impacts of PD-1 blocking/agonism seems warranted.

Minor comments:

The authors use Balb/c mice throughout, a strain with known skewed Th2 differentiation potential. A comparison with B6 mice would be interesting to determine if at least some of the critical metabolic and other findings are conserved. There are well-known strain differences in the marker expression of ILC2s between B6 and Balb/c mice (for example, reviewed in Entwistle, Front Immun, 2020).

The RNA sequencing results comparing WT and PD-1 KO ILC2s are not shown in their entirety. Suggest adding a supplemental table with all significant changes.

Please provide information within the figure legends as to what assay was used and the experimental schemes such as IL-33 administration.

In Figures 2C-F, the unit of measurement in the y axis are lacking.

In Figure 5G, the authors do not include the control Rag2KO + isotype + IL-33, to provide appropriate comparisons. In other words, does anti-PD1 contribute to heightened AHR in RAG mice? Or do RagKO mice have altered IL-33 driven AHR (with or without anti-PD1)?
Line 107 ...“the lack of PD-1 in aILC2s resulted in 840 differentially...” Similarly, line 246 “...proved that PD-1 regulates lung inflammation mainly through the regulation of ILC2s.” Suggest rephrasing, as the mice lack PD-1 in all cells, not just ILC2s. Even in the Rag KO mice, the authors cannot definitely say ILC2s are the relevant target of PD1 blockade (other ILCs, NKs, etc). Cell-intrinsic impacts of PD-1 signaling on ILC2s were not directly determined here.

Line 188 “...was more important in ILC2s lacking...” the meaning here is unclear, typo?

There is no discussion of relevant sources of PD-L1/PD-L2 that would restrict IL-33 activated ILC2s. Autocrine seems possible. Obviously data on this point would strengthen novelty of the work. In any case, suggest discussion of these important points.

We would like to thank the reviewers for their constructive remarks. We have taken their comments on board to improve and clarify the manuscript. Please find below a detailed point-by-point response to all comments.

Reviewer #1 (Remarks to the Author):

In the manuscript titled "PD-1 pathway regulates ILC2 metabolism and PD-1 agonist treatment ameliorates airway hyperreactivity" the authors investigate the previously described role of the PD-1/PD-L1 axis in negatively regulating allergic airway disease pathogenesis. Specifically, they demonstrate that ILC2 PD-1 engagement is necessary to prevent a shift in metabolism towards glycolysis (as has been shown in T cells), which was shown to be required for both expression of the master transcriptional regulator GATA3 and proliferation. Further, the authors perform *in vivo* studies demonstrating the utility of a novel PD-1 agonist in the treatment of allergic airway disease. Importantly, this therapeutic approach was tested with humanized mice and nearly abrogated airway hyperresponsiveness and inflammation, and thus represents a candidate pathway for future drug development. However, addressing several points below would significantly improve the impact of the work.

1) Lack of clinically relevant allergen. The authors utilize IL-33 throughout the study in order to model allergic airway disease pathogenesis. While the IL-33-induced response can be similar to a response-induced by a clinically relevant allergen, this is not necessarily the case. Thus, the authors should perform studies demonstrating the importance of the PD-1/PD-L1 axis with a clinically relevant allergen, such as HDM or *Alternaria*. These studies would also provide insight into how PD-1 might regulate epithelial cytokines.

We agree with the reviewer that clinically relevant allergens such as HDM or *Alternaria* are important to support our observations. To address this point, additional experiments were performed and added in our revised manuscript. In response to *Alternaria*, PD-1 expression was strongly induced on pulmonary ILC2s (Sup Fig 3F). Furthermore, PD-1 blockade in Rag2^{-/-} mice challenged with *Alternaria*, resulted in a significant exacerbation of AHR and lung inflammation (Sup Fig 3). We also confirmed the efficiency of PD-1 agonist in a previously established HDM-induced humanized asthma model¹. Briefly, preventive and therapeutic protocols were implemented in a clinically relevant asthma model relying on Rag2^{-/-} Il2rg^{-/-} mice reconstitution with total PBMCs from HDM positive patients, as presented in Sup Fig 4 C,F. Interestingly, both protocols indicated that PD-1 agonist abrogates AHR and eosinophil recruitment (Sup Fig 4D, E, G, H). Altogether the results from murine and humanized mice models support the notion that PD-1 agonist therapy alleviates symptoms associated with allergic AHR.

2) Lack of mechanism for PD-1 regulation of ILC2 responses through metabolic changes *in vivo*. Though the association with metabolic changes in ILC2s regulated by PD-1 are interesting, it is not clear that PD-1 regulation of ILC2 airway responses and the asthma phenotype *in vivo* occurs through these ILC2 metabolic changes.

In this study we implemented different and complementary approaches to assess metabolic changes in ILC2s, including RNAseq, glucose uptake and receptor experiment, metabolic flux analysis and metabolomic quantification. All those experiments suggest that lack of PD-1 was mainly associated with a remarkable increase in glucose uptake and glycolysis. To address the reviewer comments and assess the role of metabolic changes *in vivo*, we designed new experiments and the results are now added as Fig 5G-K. Briefly, WT and PD-1^{-/-} mice were injected with glycolytic inhibitor 2-deoxyglucose (2-DG) intraperitoneally along with intranasal IL-33 administration to induce AHR. Interestingly, inhibition of glycolysis, significantly decreased lung ILC2 number and proliferation in PD-1^{-/-} mice, while pulmonary ILC2 number in WT mice treated with either vehicle or 2-DG were comparable. This suggests that PD-1 prevents a metabolic shift towards glycolysis in activated ILC2s, explaining their high proliferative capacity in PD-1 deficient mice. Altogether, these new results confirm that PD-1 is a potent metabolic checkpoint in ILC2s restricting their activation and proliferation during allergic asthma.

3) Unknown effect of PD-1 agonism on non-ILC2s. The authors demonstrate that the PD-1 agonist nearly abrogates allergic airway disease pathogenesis in a humanized mouse model. However, given that the agonist only targets human PD-1 and that the only human PD-1 in this system is expressed by the human ILC2s, how would mouse PD-1 agonism perform *in vivo* in a non-humanized model? Would disease still be abrogated? This is important to understand if we are to administer PD-1 agonists in humans which have other PD-1 expressing cells in addition to ILC2s. Additionally, histological staining for inflammation and mucous hypersecretion would help readers assess the utility of PD-1 agonism in the treatment of disease.

We understand the reviewer's concern. As mentioned above, this point was addressed in Rag2^{-/-} Il2rg^{-/-} mice reconstituted with total PBMCs from HDM-allergic patients. Interestingly, PD-1 agonist significantly alleviated phenotypes associated with AHR, such as lung resistance and BAL eosinophils (**Sup Fig 4C-H**). These results support the notion that PD-1 agonist is able to down modulate PD-1⁺ ILC2s and non ILC2s in asthma. We also performed lung histology and quantify the inflammation by measuring the infiltrating cells and thickening of epithelial cells. The results clearly suggest that PD-1 therapy significantly decreased inflammation in the lungs. We now added the histology results as **Fig 7K-M**.

4) Unknown endogenous PD-L1 signal. The authors demonstrate the necessity and sufficiency of the PD-1/PD-L1 axis in dampening allergic airway inflammation; however, they do not show which cells are the physiologic source of PD-L1 in the asthma model.

This is a great question. We carefully looked at PD-L1 and PD-L2 expression in the lungs. Overall, we observed an increase in PD-L1 and PD-L2 expressions in total lung cells after IL-33 administration. Next we specifically gated on various subpopulations and our flow cytometry results clearly suggest that Gr1⁺ cells represent the majority of PD-L1⁺ cells and CD11c⁺ populations represent the majority of PD-L1/L2 double positive cells. These results were added in **Figure 1E-J** and we discussed the expression of PD-1 ligands in our manuscript (**line 323-333**). It is important to mention that our group and others previously reported the inducibility of PD-L1 and PD-L2 expression on lung dendritic cells and macrophages^{2,3}. Although lungs can provide a heterogeneous source of PD-1 ligands, PD-1 crosslinking in ILC2s is dictated by anatomical proximity. Recently, micro-anatomic niches around lung bronchi were identified as sites for colocalization and possible interaction between ILC2s and DCs⁴. This may suggest that DCs provide the major immune source of PD-L1/PD-L2 to ILC2s in the lungs. Since PD-1 and PD-L1 are both expressed by ILC2s, there is also a clear possibility of direct interactions within the same population. This possibility was also explored and addressed in response to reviewers comment below and is now incorporated in the revised manuscript as **Figures 2H and 2I**.

Minor comments:

- 1) A few confusing sentences and grammatical errors. I mention a couple comments regarding this issue in my below minor comments, however, I suggest that the authors carefully proofread and edit the entire manuscript again in order to make the manuscript as easy to follow as possible.
- 2) Line 115: Suggest changing to "...suggesting a decrease in controlled...".
- 3) Line 145: Suggest removing "did".
- 4) Line 154: Suggest changing "improved" to "altered".
- 5) Line 165: Suggest changing "utilization" to "metabolism".
- 6) Line 188: Suggest changing "important" to something more appropriate.
- 7) Line 245/341: Suggest changing "proved" to something more appropriate.
- 8) Line 260: Suggest changing "sensitized" to "challenged".
- 9) Line 274-275: Suggest rephrasing.
- 10) Line 284-88: These sentences do not appear to be clearly related to the work's findings.
- 11) Line 317: Do the authors mean to say these are not differentially expressed? Please clarify.
- 12) Line 319: This is a bit confusing as ILC2 PD-1 expression is also context dependent (e.g. IL-33 induced). Suggest removing "in which the involvement of PD-1 in Th1 and Th2 polarization is context dependent".
- 13) Line 340: Suggest changing to independent of B/T cells.
- 14) Line 665: Suggest removing 'in'

We thank the reviewer and edited all the above mentioned phrases and sentences and also proofread the entire manuscript.

Reviewer #2 (Remarks to the Author):

Using mouse models, this manuscript by Helou et al reports two main relevant findings: (1) The inhibitory immune receptor PD-1 restrains the proliferation and effector function of group 2 innate lymphoid cells (ILC2s) in allergic airway inflammation by acting on ILC2 metabolism. (2) PD-1 agonism is tested as a potential new way to treat asthma. The study is based on well-designed experiments and the central conclusions are supported by the data. While the finding that PD-1 inhibits ILC2 function is not novel, the authors provide a new mechanism of how this occurs, i.e. through preventing the metabolic reprogramming that is necessary for ILC growth and cytokine production. Investigation of ILC metabolism is a topical area and the findings reported in the manuscript are of interest to the readership of Nature Communications. To further strengthen the study, the following comments should be addressed.

Major comments

(1) Figure 1 (**Figure 2**) and Figure 4 (**Figure 5**) show that PD-1 limits the survival and proliferation of activated ILC2s, respectively. It is unclear whether the enhanced proliferation of PD-1 knockout ILC2s is due to better survival or due to a selective effect of PD-1 on proliferation. Cell death and proliferation should therefore be measured simultaneously in wild-type and PD-1-deficient ILC2s

The proliferation results were confirmed using a fixable Live dead added before the fixation/permeabilization step. We now provided the full gating strategy as **Sup Fig 1A**. In addition, the new set of *in vivo* experiments confirms that the enhanced proliferation of PD-1^{-/-} ILC2s is mainly associated with the metabolic regulation (**Fig 5G-K**).

(2) The authors show that activated ILC2s express both PD-1 and its ligand PD-L1 (Figure 1). Does PD-1 regulate ILC2 function in cis (through PD-L1 on ILC2s) or in trans (PD-1 ligands provided by other cells)?

We designed several experiments to address this important point. Briefly, lung ILC2s from naïve WT and PD-1^{-/-} mice were sorted and activated with IL-33 *in vitro*. Interestingly, in the absence of any other source of PD-L1, ILC2s from WT mice displayed different activation and response to IL-33 as compared to ILC2s from PD-1^{-/-} mice. In particular, WT ILC2s displayed a decreased expression of GATA-3 and ki67, lower production of IL-5 and a decreased cell death (**Fig 2 H, I** and **Sup Fig 2A-D**). This indicates that a potential cis interaction between PD-1 and PD-L1 could downregulate WT ILC2 activation in comparison to PD-1^{-/-} ILC2s. To assess this potential interaction further, we also performed additional experiments utilizing anti PD-1 blocking antibody. In this experiment, WT ILC2s were cultured *in vitro* in the presence of anti-PD-1 or isotype control. Interestingly, PD-1 blockade led to a significant increase in GATA-3 expression and IL-5 secretion (**Fig 2H, I**). It is important to note that while these results suggest that cis interactions may occur, certainly we cannot exclude the possibility of trans interactions with local cells. Interestingly, we addressed the similar interactions between ICOS and ICOS-L on ILC2s and reported it in *Immunity* in 2015⁵.

(3) Related to Figure 5 (**Figure 6**): Does PD-1 play an ILC2-intrinsic role in regulating IL-33-induced allergic airway inflammation? The authors exclude TH2 (and B) cells by performing experiments in Rag2 knockout mice, but other PD-1-expressing cells, such as alveolar macrophages, could be involved.

We agree with the reviewer that other pulmonary cells do express PD-1 in Rag2^{-/-} mice and could be also affected by PD-1 blockade in our approaches. We believe the result of our experiments in humanized mice clearly demonstrate that targeting PD-1 on ILC2s is enough to abrogate AHR and lung inflammation. In these experiments anti PD-1-agonist could only act on human ILC2s. This suggests that PD-1 plays a potent inhibitory role in ILC2s during the IL-33-dependent inflammatory context, without excluding a potential role for PD-1 on other immune cells. Furthermore, we now included the result of humanized mice with human PBMC and HDM challenge, which clearly suggest that PD-1 agonist can abrogate inflammatory cells causing lung inflammation (**Sup Fig 4C-H**).

(4) Figure 5E (**Figure 6E**) shows that, in steady-state, the number of lung ILC2s is increased in PD-1 knockout mice. However, the authors report that naïve lung ILC2s hardly express PD-1 surface protein (Figure 1B). The authors should reconcile these observations. Taylor et al. J Exp Med 2017 published that lung ILC2s are increased in mice lacking PD-1, but also demonstrate that ~10% of lung ILC2s express surface PD-1 in steady-state.

We agree with the reviewer that the possible expression of PD-1 on naïve ILC2s may play an important role in ILC2 homeostasis and we should certainly consider this possibility. We now clarify this issue by adding the frequency of PD-1⁺ ILC2s from naïve and IL-33 stimulated mice (**Sup Fig 1B**) and highlighted this

information in the revised manuscript (**line 103-105**). In summary, in line with the results of Taylor *et al.*, we showed that the percentage of pulmonary PD-1+ ILC2s is approximately 10%.

(5) In contrast to mouse lung ILC2s (Figure 1B), human blood ILC2s express PD-1 constitutively, i.e. in the absence of IL-33 treatment (Figure 6B) (**Figure 7B**). Is this a species-specific difference or is PD-1 induced by IL-2 and IL-7 that were used to culture human ILC2s *in vitro*? This is a relevant point since, in humans, PD-1 agonism might have unwanted effects on ILC2s in the absence of airway inflammation.

Although human peripheral blood ILC2s are not considered mature at steady state, we fully agree with the reviewer about the necessity of showing PD-1 expression before culture with IL-2 and IL-7. We now performed additional experiments and assessed the expression of PD-1 on freshly isolated ILC2s from PBMCs of healthy individuals. The results suggest that ILC2s do not express PD-1 at the steady state (**Sup Fig 4A**). This suggests that certainly as the reviewer wisely suggested, IL-2 and IL-7 additions *in vitro* are able to induce PD-1 and undoubtedly IL-33 is robustly able to upregulate PD-1 expression further. This information clearly suggests that PD-1 agonistic treatment is only able to target activated ILC2s and do not affect circulating ILC2s in the periphery.

(6) Related to Figure 6 (**Figure 7**): Is PD-1 agonist treatment also effective in ameliorating ongoing allergic airway inflammation? This is relevant for possible asthma treatment in humans. To address this question, mice should be treated with PD-1 agonist at later time, i.e. after IL-33-induced airway inflammation.

We addressed this important point by designing new therapeutic approaches, utilizing PD-1 agonist on humanized mice and after establishment of airway inflammation. Briefly, a therapeutic protocol was implemented in a clinically relevant asthma model relying on Rag2^{-/-} Il2rg^{-/-} mice reconstitution with total PBMCs from HDM positive patients, as described in **Sup Fig 4F**. Interestingly, the result of those experiments indicate that PD-1 agonist treatment abrogates eosinophil recruitment and significantly ameliorates ongoing AHR (**Sup Fig 4 G, H**) supporting our claim that PD-1 agonist may serve as a potential therapeutic target for the treatment of patients with asthma.

(7) Does IL-33 induce airway hyper-reactivity in Rag2 Il2rg double knockout mice in the absence of transferred human ILC2s (Figure 6) (**Figure 7**)? In other words, is airway hyper-reactivity driven by the transferred human ILC2s or by residual IL-33-responsive mouse cells in Rag2 Il2rg double knockout mice?

We understand the reviewer's concern. As previously demonstrated by our team, Rag2^{-/-} Il2rg^{-/-} mice do not develop AHR in the absence of transferred ILC2s with or without IL-33 treatment⁶ (Galle-Treger *et al.*, Figure 4C). We also performed these experiments and added in the manuscript as **Sup Fig 4B**.

Minor comments

(1) ILCs and T cells are related, and many concepts established for T cells apply to ILC biology. However, ILCs also possess some unique features. Therefore, the similarities as well as potential differences between ILC2 and Th2 metabolism should be discussed.

As requested, we now included differences and similarities in the revised manuscript (**line 334-342**).

(2) Statistics: ANOVA (not Student's t-test) should be used for multi-group comparisons in Figures 4E-F, 5D-F, 5J-K, and 6H.

We consulted with our biostatistician at USC and now used ANOVA and Bonferroni post hoc test for multigroup comparisons.

(3) It is unclear why isotype-treated wild-type mice are used as controls for anti-PD-1-treated Rag2 knockout mice in Figure 5G-K (**Figure 6G-K**). It seems more appropriate to use isotype treated Rag2 knockout mice instead.

We apologize for this negligence. The 4 groups of mice are Rag2^{-/-} as indicated in the scheme (**Fig 6G**) and in the figure description. The legend was corrected accordingly.

(4) Related to Figure 6H (**Figure 7I**): Representative flow cytometry plots of human lung ILC2s should be shown in control and PD-1 agonist-treated mice.

A representative flow cytometry plot showing human ILC2s in mice lungs was added (**Fig 7H**). Given the fact that ILC2s represent the only human population in our humanized mouse model, the percentages of CRTH-2⁺ CD127⁺ cells (from human CD45⁺ Lineage⁻ cells) are consistently comparable in the control and PD-1 agonist-treated mice.

(5) The legend of Figure 2 states n=6, whereas only 3-4 data points are shown in the figure.

The reviewer is correct as we only performed experiments on 3-4 mice. We corrected the appropriate legends.

(6) The legend for Figure 4 panels K and L is missing.

Thank you. We now added the legends in the revised manuscript.

(7) In the methods section, information about antibody clones and concentrations should be added to facilitate the ability of other researchers to reproduce the work.

This information is now added as requested, however, we need to point out that this information will be included as "reporting summary" according to the Nature communications editorial policy.

Reviewer #3 (Remarks to the Author):

The manuscript by Helou et al describes the role of PD-1 in regulating ILC-2 metabolism in airway hyperreactivity. The authors demonstrate that PD-1 signaling can block essential proliferative signalling pathways such as STAT5, apoptotic pathways and can modulate the metabolic profile of ILCs. The authors then explore the possibility that the increased proliferative potential of PD1 deficient ILC2s is due to differential metabolic capacity and show that blocking glucose (2DG) or methionine metabolism (cycloleucine) directly correlated to regulating ILC-2 proliferation.

The study employs significant amount of transcriptomics and metabolomics to define the mechanistic role of PD-1 in ILC-2 biology. It confirms previous observations that PD-1 (1) can inhibit ILC2 proliferation and (2) can inhibit ILC2 cytokine production (ref. 38).

The authors further define two new function for PD-1 in ILC-2s through transcriptomics data which include (a) regulation of apoptosis and (b) regulation of glycolysis.

The most notable and novel finding within this work is the use of PD-1 agonist as a therapeutic target for treating human ILC2 mediated AHR. There are several questions pertaining to the study which requires further clarifications.

While the role of PD-1 in ILC-2s have been previously shown, here the novelty clearly lies in the fact that PD-1 can regulate an apoptotic signature in ILC2s. Although the Annexin data is a clear functional test, whether the anti-apoptotic phenotype is a by-product of increased STAT5 signaling in response to IL2/IL7 is not clear. For instance, can the authors clarify if the anti-apoptotic phenotype seen can be uncoupled from the STAT5 differences seen in transcriptomics data presented here.

We agree with the reviewer that the anti-apoptotic phenotype in PD-1^{-/-} ILC2s could be related to the increased STAT5 signaling, previously demonstrated by Taylor *et al.*⁷ However, this phenotype could also be related to the altered SHP2-mediated regulation of PI3K/Akt/mTor pathway, independently from STAT5 signaling^{8,9}. Additionally, our new results showed that the lack of PD-1 significantly increases the expression of the major anti-apoptotic factor Bcl-2 (**Fig 2N, O**). Altogether, these information suggest that a well-maintained balance between PI3K/AKT activation, Bcl-2 expression and STAT5 signaling, could control ILC2 viability, as previously demonstrated for T cells¹⁰.

The second interesting observation is the metabolic regulation of ILC-2s. However, the functional sea horse data does not provide a clear and in-depth analysis of the glycolytic capacity of these cells, thereby making it difficult to prove that PD-1 has an effect on ILC-2 metabolism.

We agree with the reviewer that Seahorse data is often not enough to draw conclusions regarding cellular metabolism. We need to point out that we designed several complementary approaches to validate and understand ILC2 metabolic activity. First, we carefully assessed transcriptomic data by analyzing several glycolysis-related genes in PD-1^{-/-} ILC2s and compare the results to the WT controls (**Fig 3A**). Second, we performed glucose uptake experiments and assessed the level of Glut-1 expression by flow cytometry (**Fig 3B and Sup Fig 2B**). Third, we analyzed our samples with a high-resolution mass spectrometry system (HRMS) coupled with liquid chromatography (UHPLC) and direct infusion (NanoMate) for separation and detection of various classes of small molecules and metabolites (**Fig 3C-F**). Overall, these results along with the results of Seahorse suggest that lack of PD-1 on activated ILC2s, significantly induce glycolysis.

Furthermore, we now designed new approaches and performed *in vivo* experiments with glycolysis inhibitor 2-DG. Interestingly, inhibition of glucose pathway significantly decreased ILC2 number and proliferation in PD-1^{-/-} mice but does not affect the ILC2s in WT mice. This suggests that PD-1 prevents a metabolic shift towards glycolysis in activated ILC2s, explaining their high proliferative capacity in PD-1 deficient mice. Altogether, these new results confirm that PD-1 is a potent metabolic checkpoint in ILC2s restricting their activation and proliferation during allergic asthma. The new results are now added as **Figure 5G-K**.

The third major observation within the manuscript is the use of PD-1 agonists in AHR. A similar experimental murine model was performed in the Yu et al, Nature, 2016, Vol.539, 102-106; where the authors showed the opposite effect with papain challenge and influenza challenge. In both these models, it was shown that PD-1 blocking antibodies decreased ILC2 cytokine production leading to less eosinophil accumulation. Can the authors discuss the discrepancies between the two studies?

We thank the reviewer for this interesting remark. Yu *et al.*, have injected a PD-1 antibody (clone J43), which was reported to reduce mouse CD4⁺PD-1⁺ T-cell numbers by complement-dependent cytotoxicity¹¹. The authors performed repeated administration of PD-1 antagonist and claimed that the J43 clone specifically depleted PD-1hi and IL-5-producing ILC2s. According to the authors description, PD-1 was not blocked on ILC2s but PD-1⁺ ILC2s were depleted. In our study, we injected mice with a single dose of a different clone of PD-1 antagonist (clone 29F.1A12; BioXcell) that was only reported as a blocking antibody (no depletion). Therefore, there are no real discrepancies between the two studies: depletion of PD-1hi⁺ ILC2s logically leads to less cytokine production and eosinophil accumulation while blocking PD-1 on ILC2s increases cytokine production, proliferation and exacerbates AHR.

Figure 1.

The characterization of ILC2s as naïve, mature and inflammatory needs to be clearly defined. Was a live/dead gate utilised in the analysis and if yes it would be good to add the gating strategy. In plots H&I (**Fig 2L, M**), the data shown are from n=4 mice, but the legend describes this as a cumulative data of 3 experiments. Is this representative or cumulative data? While it is hard to perform western blotting with such limited cell numbers, would it be possible to confirm any of the transcriptomics data with protein data for genes shown in Fig.1F (**Fig 2J**)?

We thank the reviewer for these constructive comments.

- In our study, naïve ILC2s refer to ILC2s from naïve mice that were not challenged with IL-33. In parallel, activated ILC2s refer to ILC2s from mice that were challenged intranasally with IL-33. The figure legend section was carefully edited and corrected in the revised manuscript. We need to point out that we did not use any markers such as KLRG to discriminate mature and immature ILC2s.
- Gating strategy with Life/dead staining is now added to the study **Sup Fig 1A**. Moreover, for the apoptosis experiments DAPI was used to discriminate late apoptotic and necrotic cells.
- We confirm that our graphs in Figure 2L, M show representative data.
- Unfortunately, it is not technically feasible to perform western blot as the ILC2 yield from each mouse is relatively low. However, as suggested, we now confirmed our previous results by assessing protein level of anti-apoptotic factor Bcl2 by flow cytometry. In line with our results, Bcl2 expression was higher in activated PD-1^{-/-} ILC2s compared to WT control, confirming the role of PD-1 in ILC2 survival. These results are now added as **Figures 2 N, O**.

Figure 2 (**Figure 3**)

The authors demonstrate that PD-1 regulates glycolysis but the data shown in Fig. 1 does not optimally demonstrate this phenomenon. The sea horse analysis simply shows mitochondrial respiration. It would be more appropriate to measure ECAR with glucose, oligomycin and 2DG? Further, the OCR/ECAR ratio is at maximal respiration (assuming after FCCP injection), what is the ratio for basal respiration? Identifying basal glycolytic rate, maximal glycolytic rate and glycolytic capacity may provide more insight into the role of PD-1 in glycolysis? Also, can the authors clarify what the y axis scale denotes in panels c-f? Is it fold change?

Functional seahorse experiments require large quantity of cells (>1 million ILC2s per sample) and numerous mice of each genotype. As ILC2s were previously described to prefer FAO and uptake lipids *in vitro* and *in vivo*¹²⁻¹⁴, our seahorse characterization initially focused on OCR data and use of this pathway with Etomoxir, while still providing ECAR phenotype. We agree that a glycolytic rate assay would be an ideal follow-up but would take a larger number of animals and reagents which is currently not available. However, we communicated with the metabolic core and reanalyzed our data and now provide a new graph showing cell energy phenotype, as previously described¹⁴. This new representation simultaneously represents the increased ECAR in PD-1^{-/-} ILC2s as well as the OCR/ECAR ratio at basal and maximal respiration (**Fig 3H**). Furthermore, similar OCR/ECAR ratios were observed for the basal respiration, while higher OCR/ECAR ratios were observed in WT ILC2s at maximal respiration. The glycolytic capacity was calculated as the difference between maximal ECAR after FCCP injection and basal ECAR (**Fig 3I**).

Finally, we now added the missing information in panels c-f. The Y axis does represent the relative levels of metabolites as indicated in the figure legend.

Figure 3 (**Figure 4**)

Please denote what the values on Y axis denotes in c-l.

We apologize for the missing information. The Y axis represents the relative level of metabolites as was indicated in the figure legend.

Figure 4 (Figure 2)

Can the authors clarify the use of PDL2 FC over PDL1 FC?

PD-L2 was used since it binds to PD-1 with 3-fold stronger affinity compared to PD-L1^{15,16}. In addition, PD-L1 is already expressed on ILC2s and could directly engage ILC2 PD-1, as suggested in Fig 2 H, I. This was clarified in the revised manuscript.

Minor Comments

The introductory paragraph does not fully acknowledge some of the original literature on PD-1 signaling. For instance, the work by Okazaki et al and Chemnitz et al are one of the first to demonstrate that PD-1 cytoplasmic tail recruits SHP2. Recently, further work by Ronald Vale and Rafi Ahmed's group have clarified this signalling cascade. The PD-1 stop signal work was first demonstrated by Fife et al. Also, ref. 15, 16 are not the first observations on PD-1 expression on ILC-2s. These were first demonstrated by refs. 33, 38 and Seillet et al, Cell reports, 2016, 17 (2):436-447.

We thank the reviewer for the suggestions. The references were added in the revised manuscript.

Reviewer #4 (Remarks to the Author):

Review for Helou, Akbari et al.:

Summary: The authors identify the checkpoint molecule PD-1 as an IL-33 driven activation marker of lung type 2 innate lymphoid cells (ILC2s) that negatively regulates ILC2 proliferation, survival, and production of type 2 cytokines within the lung. The authors show that PD-1 is upregulated in IL-33 activated lung ILC2s. In PD-1 deficient mice, ILC2s can proliferate, survive, and produce more cytokines in response to IL-33, independent of T and B cells. The authors demonstrate that a lack of PD-1 induces/heightens a switch in ILC2s to aerobic glycolysis and amino acid catabolism. The authors use a humanized mouse model to demonstrate that PD-1 agonism is sufficient to limit human ILC2s and allergic airway response in vivo.

Together, the work is well-written and the data are straightforward and convincing. This study is the first to suggest a metabolic mechanism behind PD-1 regulation in pulmonary ILC2s, potentially highlighting PD-1 agonism as a therapeutic potential for the control of allergic asthma. One drawback is the prior publications that cover similar concepts (ILC2s and PD-1 signaling). For example, Taylor, JEM 2017, which shows PD-1 KO ILC2s have enhanced STAT5 signaling, increased proliferation and type 2 cytokine production, and offer superior protection against helminth infection. Yu, Liu, Nature, 2016 found PD-1 was highly express on ILC progenitors, but also show that activated lung ILC2s express high levels of PD-1. Moral, Nature, 2020 recently found that ILC2s in pancreatic cancer express PD-1 in an IL-33 dependent manner, restricting ILC2 function and limiting anti-tumor immunity. Together, these published works demonstrate that IL-33 can activate ILC2s to upregulate PD-1, that PD-1 signaling limits ILC2 proliferation and cytokine production, and that loss of PD1 in ILC2s (or PD-1 blockade) leads to increased ILC2 functionality/activity. Several elements of this current work overlap with these ideas, but there are also very interesting and novel points, including (1) the metabolic impact of PD-1 loss on ILC2s and (2) the potential ability to agonize PD-1 to restrict ILC2 function in the treatment of allergic asthma. Major and minor comments are listed below:

Major comments:

1) The metabolic analysis and metabolomics are clearly a strength of this work. However, aspects of the results are difficult to definitively interpret and put into context.

In our manuscript, we used important complementary approaches (including RNAseq analysis, glucose uptake test, metabolic flux analysis and metabolomic quantification) and based on the results we highlighted the critical role of PD-1 in modulating ILC2 metabolism *ex vivo*. To put our results into context and check whether the exacerbated asthma phenotype in PD-1 deficient mice occurs through the metabolic regulation of ILC2s, we performed a new set of *in vivo* experiments as described above and now added to the revised manuscript as Figure 5G-K. Our results support our previous findings and suggest that glucose inhibition significantly decreased lung ILC2 number and proliferation in PD-1^{-/-} mice but does not affect the ILC2s in WT mice. This suggests that PD-1 restricts glucose metabolism in activated ILC2s, explaining their high proliferative capacity in PD-1 deficient mice. Altogether, these new results confirm that PD-1 is a potent metabolic checkpoint in ILC2s restricting their activation and proliferation during allergic asthma.

How do these metabolic findings (Figures 2-4) (**Figures 3-5**) compare to that of naïve, non-IL-33 treated ILC2s?

We addressed this question using freshly sorted naïve ILC2s. As the number of naïve ILC2s is very limited for a full metabolomic study, we compared glucose uptake and Glut-1 expression at the steady state. Our experiments do not reveal significant differences at the level of glucose uptake. Consistent with these results, naïve WT and PD-1^{-/-} ILC2s exhibit the same expression levels of Glut-1, considered as the main transporter of glucose across the plasma membrane. Therefore, this confirms that our metabolic findings are closely related to PD-1 induction on ILC2s, required to control their activation. These results are shown in **Sup Fig 2E, F**.

How specific are these metabolic changes for lung ILC2s, for example in comparison with activated and differentiated Th2 cells?

We believe that these metabolic changes are not exclusive for pulmonary ILC2s, knowing that previous studies have demonstrated the ability of PD-1 to reprogram T cells metabolism from glycolysis to fatty acid oxidation¹⁷. Ogando et al., have recently demonstrated that PD-1 crosslinking on human CD8⁺ T cells strongly affects the mitochondrial metabolism and therefore alters glycolysis, oxidative phosphorylation (OXPHOS) and reprograms CD8⁺ T cells metabolism for FA oxidation¹⁸. These results are consistent with our study showing that PD-1 deficiency reprograms ILC2 metabolism toward glycolysis and amino acid metabolism in an IL-33 induced-asthma context. We now discussed the metabolic requirements in ILC2s and T cells and highlighted the similarities and differences in the revised manuscript (**line 334-342**).

Are these ILC2 changes unique to PD-1 inhibition, or does this PD-1 KO ILC2 profile more generally reflect the differences between a more activated and less activated lung ILC2? For example, it seems plausible that more highly activated lung ILC2s (PD1 KO ILC2s after IL-33 treatment) would be more dependent on glycolysis and amino acid catabolism, similar to other highly activated lymphocytes. These questions are meant to illustrate some of the conceptual points that could strengthen the author's conclusions and add to the novelty of the work.

We thank the reviewer for these interesting questions. As discussed above, the PD-1-mediated metabolic regulation also applies to T cells. Although it seems plausible that highly activated cells would upregulate their metabolism, the induction of glycolysis upon activation is not necessarily a defining feature of ILC2s. Several studies support the idea of a preferential dependence on fatty acid metabolism in many contexts¹²⁻¹⁴. Using the fatty acid oxidation inhibitor "Etomoxir", our lab has also recently evidenced that lipid metabolism is critical for ILC2 effector function¹⁴. In parallel, the respective inhibitors of glycolysis (2-DG) and methionine catabolism (CYL) do not affect WT ILC2 effector function in our *ex vivo* and *in vivo* experiments (2-DG) (**Fig 5**). This confirms and highlights a metabolic shift towards glycolysis and amino acid catabolism in PD-1 deficient ILC2s. We discussed further this conceptual point and illustrated it better in the revised version.

2) The authors use their results with a PD-1 agonist in humanized mice to infer possible pathways within asthma patients. However, their conclusions would be strengthened by comparing their results with a better-established model of allergic asthma (e.g. House dust mite, papain, fungal infection, chitin etc...) to determine the impact of PD-1 signaling. Although IL-33 administration is a clean system, its physiologic relevance is less clear.

IL-33 model was used as a proof of concept. Nevertheless, we agree with the reviewer about the importance of a clinically relevant allergen to strengthen our observations. As discussed above, we confirmed the efficiency of PD-1 agonist in a previously established HDM-induced humanized asthma model¹. Briefly, preventive and therapeutic protocols were implemented in a clinically relevant asthma model relying on Rag2^{-/-} Il2rg^{-/-} mice reconstitution with total PBMCs from HDM positive patients, as described in **Sup Fig 4 C,F**. Interestingly, both protocols indicated that PD-1 agonist abrogates AHR and eosinophil recruitment (**Sup Fig 4D, E, G, H**). Altogether these results from IL-33 and HDM-mediated humanized mice asthma models support the potential of PD-1 agonist therapy in improving allergic asthma.

3) In Figure 5D (**Figure 6 E, F**), there seems to be a difference in number and cytokine expression between naïve PD-1KO and WT ILC2s, suggesting that PD-1 has impacts on ILC2s (or other cells that indirectly impact ILC2s) at steady state. Therefore, a complicating factor in the difference between IL-33 treated WT and PD-1KO ILC2s is their baseline activation state. To address these concerns, the authors could provide

further characterization of PD-1KO mice at steady state. In a related point, the contribution of ILC2-intrinsic PD-1 signaling, versus signaling on other cellular targets, is not well-established in this work. Some effort to determine direct (ILC2-intrinsic) versus indirect impacts of PD-1 blocking/agonism seems warranted.

We understand the reviewer's concern. The characterization of PD-1^{-/-} mice at steady state has been fairly addressed previously by Taylor *et al.* Briefly, their work showed that WT and PD-1^{-/-} ILC2s display similar expression of IL-7R and IL-2R. Although the number of ILC2s was higher in PD-1 deficient mice in different organs including the lungs, the mixed chimera experiments revealed that PD-1 does not affect ILC2 development suggesting an intrinsic regulation of ILC2 number⁷.

Additionally, as mentioned above, ~10% of ILC2s express PD-1 at the steady state. Therefore, we could not exclude an effect of PD-1 in this subset at the steady state and consequently a higher number of naïve ILC2s in the lungs of PD-1^{-/-} mice compared to WT control. It is worthwhile mentioning that the increased number of IL-5 IL13⁺ ILC2s in PD-1^{-/-} mice (**Fig 6F**) is related to the increased number of ILC2s in these mice compared to WT mice (**Fig 6E**), as the percentage of IL-5 IL-13⁺ ILC2s was the same in both genotypes.

Since our study aims to uncover the implication of PD-1 in ILC2 metabolism, we compared glucose uptake and Glut-1 expression at steady state as mentioned above (**Sup Fig 2E, F**) and observed no statistical differences, suggesting that PD-1 regulates ILC2 metabolism upon activation.

Minor comments:

The authors use Balb/c mice throughout, a strain with known skewed Th2 differentiation potential. A comparison with B6 mice would be interesting to determine if at least some of the critical metabolic and other findings are conserved. There are well-known strain differences in the marker expression of ILC2s between B6 and Balb/c mice (for example, reviewed in Entwistle, Front Immun, 2020).

We understand the reviewer's concern. Our PD-1^{-/-} mice are backcrossed to the BALB/c background in Harvard several years ago as this is a preferential genetic background to induce lung inflammation. Unfortunately, due to the unfortunate situation right now along with limited time given by the editors, we cannot perform experiments with B6 background. However, we should emphasize that many aspects of our study are in line with the results reported with Taylor *et al.* -using PD-1^{-/-} mice on a B6 background.

The RNA sequencing results comparing WT and PD-1 KO ILC2s are not shown in their entirety. Suggest adding a supplemental table with all significant changes.

As per the reviewer request, we added a supplemental table showing the top 50 differentially regulated genes, and we will provide the Gene Expression Omnibus database (GEO) per journal policy.

Please provide information within the figure legends as to what assay was used and the experimental schemes such as IL-33 administration.

An experimental description was added accordingly in each figure legend.

In Figures 2C-F (**Figure 3C-F**), the unit of measurement in the y axis are lacking.

The Y axis represents the relative level of metabolites. We now added the unit in the axis in the revised manuscript.

In Figure 5G (**Figure 6G**), the authors do not include the control Rag2KO + isotype + IL-33, to provide appropriate comparisons. In other words, does anti-PD1 contribute to heightened AHR in RAG mice? Or do RagKO mice have altered IL-33 driven AHR (with or without anti-PD1)?

We apologize for this negligence. The 4 groups of mice are Rag2^{-/-} as indicated now in the scheme **Fig 6G**. However, our cumulative data in the lab showed that Rag2^{-/-} mice have generally higher IL-33 driven AHR compared to WT mice with higher number of ILC2s (**Fig 6B, E vs 6H, K**).

Line 107 ... "the lack of PD-1 in aILC2s resulted in 840 differentially..." Similarly, line 246 "...proved that PD-1 regulates lung inflammation mainly through the regulation of ILC2s." Suggest rephrasing, as the mice lack PD-1 in all cells, not just ILC2s. Even in the Rag KO mice, the authors cannot definitely say ILC2s are the relevant target of PD1 blockade (other ILCs, NKs, etc). Cell-intrinsic impacts of PD-1 signaling on ILC2s were not directly determined here.

These sentences were edited accordingly in the revised version. We agree that other ILCs, NKs are present in Rag2^{-/-} mice, however we do believe that this model helps confirming our hypothesis since ILC2s are the main responders in IL-33 induced AHR.

Line 188 "...was more important in ILC2s lacking..." the meaning here is unclear, typo?

We apologize for this typo; this was accordingly corrected in the revised version.

There is no discussion of relevant sources of PD-L1/PD-L2 that would restrict IL-33 activated ILC2s. Autocrine seems possible. Obviously, data on this point would strengthen novelty of the work. In any case, suggest discussion of these important points.

As discussed above and in response to the other reviewer, we assessed PD-L1 and PD-L2 expressions in total lung cells and carefully reported subpopulations by flow cytometry. Please see **Fig 1E-J; Fig 2H, I** and relevant results and discussion in the revised manuscript (**line 323-333**).

References:

1. Duez, C. *et al.* House Dust Mite-induced Airway Changes in hu-SCID Mice. *Am. J. Respir. Crit. Care Med.* **161**, 200–206 (2000).
2. Akbari, O. *et al.* PD-L1 and PD-L2 modulate airway inflammation and iNKT-cell-dependent airway hyperreactivity in opposing directions. *Mucosal Immunol.* **3**, 81–91 (2010).
3. Loke, P. & Allison, J. P. PD-L1 and PD-L2 are differentially regulated by Th1 and Th2 cells. *Proc. Natl. Acad. Sci. U. S. A.* **100**, 5336–5341 (2003).
4. Dahlgren, M. W. *et al.* Adventitial Stromal Cells Define Group 2 Innate Lymphoid Cell Tissue Niches. *Immunity* **50**, 707-722.e6 (2019).
5. Maazi, H. *et al.* ICOS:ICOS-Ligand interaction is required for type 2 innate lymphoid cell function, homeostasis and induction of airway hyperreactivity. *Immunity* **42**, 538–551 (2015).
6. Galle-Treger, L. *et al.* Nicotinic acetylcholine receptor agonist attenuates ILC2-dependent airway hyperreactivity. *Nat. Commun.* **7**, 1–13 (2016).
7. Taylor, S. *et al.* PD-1 regulates KLRG1+ group 2 innate lymphoid cells. *J. Exp. Med.* **214**, 1663–1678 (2017).
8. Liu, Y. *et al.* PD-1-Mediated PI3K/Akt/mTOR, Caspase 9/Caspase 3 and ERK Pathways Are Involved in Regulating the Apoptosis and Proliferation of CD4+ and CD8+ T Cells During BVDV Infection in vitro. *Front. Immunol.* **11**, (2020).
9. Zhao, R. *et al.* PD-1/PD-L1 blockade rescue exhausted CD8+ T cells in gastrointestinal stromal tumours via the PI3K/Akt/mTOR signalling pathway. *Cell Prolif.* **52**, e12571 (2019).
10. Hand, T. W. *et al.* Differential effects of STAT5 and PI3K/AKT signaling on effector and memory CD8 T-cell survival. *Proc. Natl. Acad. Sci.* **107**, 16601–16606 (2010).
11. Kasagi, S. *et al.* Anti-Programmed Cell Death 1 Antibody Reduces CD4+PD-1+ T Cells and Relieves the Lupus-Like Nephritis of NZB/W F1 Mice. *J. Immunol.* **184**, 2337–2347 (2010).
12. Robinette, M. L. *et al.* Transcriptional Programs Define Molecular Characteristics of Innate Lymphoid Cell Classes and Subsets. *Nat. Immunol.* **16**, 306–317 (2015).
13. Wilhelm, C. *et al.* Critical role of fatty acid metabolism in ILC2-mediated barrier protection during malnutrition and helminth infection. *J. Exp. Med.* **213**, 1409–1418 (2016).
14. Galle-Treger, L. *et al.* Autophagy is critical for group 2 innate lymphoid cell metabolic homeostasis and effector function. *J. Allergy Clin. Immunol.* **0**, (2019).

15. Philips, E. A. *et al.* The structural features that distinguish PD-L2 from PD-L1 emerged in placental mammals. *J. Biol. Chem.* jbc.AC119.011747 (2019) doi:10.1074/jbc.AC119.011747.
16. Latchman, Y. *et al.* PD-L2 is a second ligand for PD-1 and inhibits T cell activation. *Nat. Immunol.* **2**, 261–268 (2001).
17. Patsoukis, N. *et al.* PD-1 alters T-cell metabolic reprogramming by inhibiting glycolysis and promoting lipolysis and fatty acid oxidation. *Nat. Commun.* **6**, 6692 (2015).
18. Ogando, J. *et al.* PD-1 signaling affects cristae morphology and leads to mitochondrial dysfunction in human CD8+ T lymphocytes. *J. Immunother. Cancer* **7**, 151 (2019).

REVIEWERS' COMMENTS:

Reviewer #1 (Remarks to the Author):

The authors have addressed my comments adequately and have greatly improved the manuscript.

Reviewer #2 (Remarks to the Author):

The authors have addressed all my comments thoroughly in their rebuttal and in the revised manuscript. Overall, the study has been strengthened significantly by addressing all reviewers' comments. The manuscript is therefore now suitable for publication in Nature Communications.

Reviewer #3 (Remarks to the Author):

The authors have sufficiently addressed all my comments.

Reviewer #4 (Remarks to the Author):

I appreciate the author's responsiveness to my comments. I feel the manuscript is much improved and clarified and do not have other significant concerns.

REVIEWERS' COMMENTS:

Reviewer #1 (Remarks to the Author):

The authors have addressed my comments adequately and have greatly improved the manuscript.

Reviewer #2 (Remarks to the Author):

The authors have addressed all my comments thoroughly in their rebuttal and in the revised manuscript. Overall, the study has been strengthened significantly by addressing all reviewers' comments. The manuscript is therefore now suitable for publication in Nature Communications.

Reviewer #3 (Remarks to the Author):

The authors have sufficiently addressed all my comments.

Reviewer #4 (Remarks to the Author):

I appreciate the author's responsiveness to my comments. I feel the manuscript is much improved and clarified and do not have other significant concerns.

We thank the reviewers for their positive feedbacks.